# Estimation of Species-Scale Canopy Chlorophyll Content in Mangroves from UAV and GF-6 Data

Liangchao Deng [1,2,3], Bowei Chen [1,2,*], Min Yan [1,2], Bolin Fu [3], Zhenyu Yang [2,3,4], Bo Zhang [2,3] and Li Zhang [1,2,*]

1. Key Laboratory of Digital Earth Science, Aerospace Information Research Institute, Chinese Academy of Sciences, Beijing 100094, China; d18277754458@163.com (L.D.); yanmin@aircas.ac.cn (M.Y.)
2. International Research Center of Big Data for Sustainable Development Goals, Beijing 100094, China; yzyloloo@gmail.com (Z.Y.); zhangbo203@mails.ucas.ac.cn (B.Z.)
3. College of Geomatics and Geoinformation, Guilin University of Technology, Guilin 541006, China; fubolin@glut.edu.cn
4. School of Marine Technology and Geomatics, Jiangsu Ocean University, Lianyungang 222005, China
* Correspondence: chenbw@aircas.ac.cn (B.C.); zhangli@aircas.ac.cn (L.Z.); Tel.: +86-18672318985 (B.C.); +86-18600132968 (L.Z.)

**Abstract:** The growth of mangroves is inhibited due to environmental degradation, and changes in the growing health of mangrove forests cause changes in internal physicochemical parameters. The canopy chlorophyll content is an important indicator to monitor the health status of mangroves. We study the effective inversion data sources and methods of mangrove health indicator parameters to monitor the health of mangrove ecosystems in typical areas of Beibu Gulf, Guangxi. In this study, we evaluated the capability of UAV, GF-6 data, and machine learning regression algorithms in estimating mangrove species-scale canopy chlorophyll content (CCC). Effective measures for mangrove pest and disease pressure, *Sporobolus alterniflorus* invasion, and anthropogenic risk are also explored, which are important for mangrove conservation and restoration. (1) We obtained several feature variables by constructing a combined vegetation index, and the most sensitive band of mangrove CCC was selected by the characteristic variable evaluation, and the CCC model at the mangrove species-scale was evaluated and validated. Through variable preferences, the feature variables with the highest contribution of *Avicennia marina*, *Aegiceras corniculatum*, *Kandelia candel*, and a collection of three categories of species in the UAV data were indices of RI35, MDATT413, RI35, and NDI35. (2) Random Forest, Gradient Boosting Regression Tree, and Extreme Gradient Boosting were evaluated using the root-mean-square error and coefficient of determination accuracy metrics. Extreme Gradient Boosting regression algorithms were evaluated for accuracy. In both UAV data and GF-6, RF achieved optimal results in inverse mangrove *Aegiceras corniculatum* species CCC, with higher stability and robustness in machine learning regressors. (3) Due to the sparse distribution of *Kandelia candel* in the study area and the low spatial resolution of the images, there is an increased possibility that individual image elements contain environmental noise, such as soil. Therefore, the level of CCC can effectively reflect the health status of mangroves and further reflect the increased possibility of the study area being exposed to risks, such as degradation. The establishment of the current protected areas and restoration of degraded ecosystems are effective measures to cope with the risks of mangrove pest and disease stress, invasion of *Sporobolus alterniflorus*, and anthropogenic activities, which are important for the protection and restoration of mangroves. This study provides an important data reference and risk warning for mangrove restoration and conservation.

**Keywords:** mangrove; canopy chlorophyll content (CCC); machine learning regression (MLR); unmanned aerial vehicle (UAV) imagery; Gaofen-6 (GF-6) satellite imagery

## 1. Introduction

Mangroves are salt-tolerant woody plant communities consisting of evergreen trees and shrubs, distributed in tropical and subtropical coastal intertidal and inlet estuaries. Mangrove wetlands are one of the most important blue carbon ecosystems in the world, with many ecological values, such as preventing shoreline erosion, sequestering carbon, and maintaining biodiversity [1,2]. Compared to other terrestrial vegetation, the unique structural morphology and biological and physiological characteristics of mangroves allow them to grow in extreme environments, such as salinity, high temperatures, strong winds, high tides, high sedimentation, and anaerobic soils [3]. Despite the many ecosystem services that mangroves can provide, the health of mangrove forests is threatened by natural factors, such as strong winds, pests, diseases, fouling organisms, invasive alien species, and the overexploitation of mangrove wetlands by humans in blind pursuit of economic benefits, leading to land use conversion to aquaculture, agriculture, and urban development, and the global extent of mangrove forests is rapidly degrading [4]. The Beibu Gulf in Guangxi is a vital distribution area for mangrove forests in China. Compared with the terrestrial forest, the mangrove community in Beibu Gulf of Guangxi has a single structure and low insect diversity, which makes it difficult for pests to be effectively restrained due to the small number of natural enemies, while the introduced exotic species, such as *Sporobolus alterniflorus* and other organisms, directly endanger the ecosystem health of mangroves. Canopy chlorophyll content (CCC) is related to the vegetation light capture, photosynthesis rate, and other physiological processes, and can reflect the overall health of mangrove forests [5]. Therefore, long-time series, high accuracy, and large-scale canopy chlorophyll content mapping are important for mangrove health monitoring and risk assessment. For a long time, little attention has been paid to the health of mangroves in Beibu Gulf, Guangxi. The inversion of canopy chlorophyll content, which is an important indicator for assessing the health of mangroves, still needs to be studied in depth.

Canopy chlorophyll content (CCC) can be calculated from the product of leaf chlorophyll-a content (LCC) and leaf area index (LAI) [6,7]. Most traditional estimates of the chlorophyll content and leaf area index at the leaf level are based on ground-based measurements, and although this method is the most accurate estimation, it is often time-consuming and destructive to the plant, and it is difficult to obtain the spatial variability of LAI and LCC on a large scale [7,8]. Remote sensing technology with remote sensing image sensors mounted on airborne and satellite-based platforms is a long-time, large-scale, fast, and efficient method to directly measure vegetation LAI and chlorophyll compared to traditional ground-based measurement methods [9–11]. Extracting the chlorophyll content of the canopy from remotely sensed data is complex, and for this reason, the effects of canopy structure, background noise, and geometry should be considered [12,13]. The purpose of constructing vegetation indices (VIs) is to minimize the influence of environmental noise on the estimation of vegetation parameters through the combination of spectral bands and to improve the sensitivity of spectral features to vegetation characteristics [14,15]. However, for the inversion of vegetation species-scale parameters, the spatial resolution of the images is particularly important, and Sentinel-2 data are slightly lacking. The multispectral remote sensing data acquired by the satellite-based platform have some limitations in estimating vegetation parameters. Firstly, satellite-based high-resolution images are more costly to acquire, but still cannot meet the demand of spatial resolution of images for precision forestry; especially for the inversion of mangrove species-scale, it is slightly weak [11]. Second, satellite-based multispectral images are limited by the re-entry cycle of the mounted platform, which makes it difficult to obtain cloud-free remote sensing images at a specific time, and the Unmanned aerial vehicle (UAV) platform can theoretically make up for the limitations of the satellite-based equipment [16,17]. UAVs can effectively eliminate the influence of environmental noise, such as other vegetation species and the background, due to their very high spatial resolution and the ability to adjust the flight path according to the angle of solar incidence [11]. UAV remote sensing technology can collect a wide range of spatial information in a shorter period to improve field efficiency, while maintaining

high accuracy and spatial resolution, and UAV images have been studied as the core data for collecting vegetation biophysical and chemical parameters [11,18,19]. For example, tree species, height, canopy diameter, and above-ground biomass (AGB) forest survey data acquired by UAV with the motion structure and multi-view stereo photogrammetry program (UAV-SfM) can complement and eventually replace traditional forest survey techniques [20–23]. UAV multispectral imagery has been demonstrated to be highly reliable in estimating vegetation biochemical parameters [10,24–26]. The Gaofen-6 (GF-6) satellite data, a Chinese first Gaofen satellite for precision agricultural observation, provide reflectance information sensitive to vegetation parameters. GF-6 data was used, for example, as a practical input to PROSAIL models and different machine learning regression algorithms for simulating spectral reflectance when inverting (it means estimation and retrieval) vegetation LAI [27]. The GF-6 multispectral data with spatial resolution up to 2 m and a wavelength range consistent with UAV data can be used as a satellite-based data source for species-scale parameter inversion of mangroves. At present, the reliability of using UAV and GF-6 multispectral data as a data source for the inversion of mangrove species-scale canopy chlorophyll content still needs to be further explored [28].

Currently, empirical remote sensing estimates of vegetation biophysical parameters are mainly divided into parametric and non-parametric regression methods. The reflectance in the red-edge region (wavelengths in the range of 680–750 nm) is considered the best remote sensing characterization indicator of chlorophyll concentration, and therefore, the direct statistical relationship between vegetation chemistry and specific reflectance, i.e., the parametric regression relationship, can be established using remote sensing measurements to model the spatial variation of the chlorophyll concentration on a large scale [9,29]. Empirical modeling approaches often establish relational expressions by comparing vegetation-specific spectral features and selecting the best vegetation index with vegetation parameters, and this approach has been successfully applied to various vegetation canopies [7,30–34]. Among the empirical statistical regression models, vegetation index parameter regression models are widely used. For example, a high correlation between the vegetation index and chlorophyll was observed in crop development stage monitoring, with spectral indices showing high sensitivity to chlorophyll content in forest vegetation at both the leaf ($R^2 = 0.72$; $p < 0.001$) and canopy ($R^2 = 0.78$; $p < 0.001$) levels [35,36]. Compared to parametric regression methods based on vegetation indices, red-edge locations, or spectral integrals, machine learning regression (MLR) algorithms produce adaptive and robust relationships that can handle the strong nonlinear relationship between biophysical parameters and observed reflected radiation in a function-dependent manner [37,38]. Machine learning regression algorithms are widely used in the estimation of vegetation parameters, such as chlorophyll concentration, due to their high assessment accuracy and stability [39–46]. For the estimation of vegetation parameters, various MLR algorithms have been shown to be more effective than parametric methods with linear models. The Random Forest (RF), Gradient Boosting Regression Tree (GBRT), and Extreme Gradient Boosting (XGBoost) algorithms are common machine learning regression algorithms and have been shown to be beneficial tools for estimating vegetation parameters [9,41,47,48]. Machine learning regression algorithms have the potential to invert mangrove CCC with stronger model generalization relative to general parametric regression methods; however, the applicability of RF, GBRT, and XGBoost algorithms in mangrove CCC estimation, especially at the species-scale, is still very limited.

The current studies for mangrove canopy chlorophyll content, especially the application of UAV data and GF-6 satellite data at the species-scale, are relatively few. From the perspective of spatial and aerial platforms, the selection of UAV and GF-6 data can balance cost and efficiency for decision-makers. The usefulness of different machine learning algorithms in estimating canopy chlorophyll content under different circumstances (different data, different species-scale) is dissimilar. The current applications of machine learning algorithms on different remote sensing data still need to be further explored. By comparing

the performance of distinct machine learning regressors, the best inversion model can be selected to improve the inversion accuracy of mangrove CCC.

With the use of mangrove UAV and GF-6 multispectral images, the study in this paper aims to construct a species-scale CCC model of mangroves using the MLR technique. The objectives of this study were (1) to investigate the capability of UAV and GF-6 multispectral data for inversion of mangrove canopy chlorophyll content in the mangroves; (2) to identify the bands and vegetation indices in UAV and GF-6 data that are sensitive to mangrove CCC; (3) to compare the accuracy of UAV data and GF-6 remote sensing images for inversion of three different mangrove species; (4) to quantify the optimal machine learning model in this area and using this method to guide the future mangrove health condition and monitoring, for conversion and restoration. This study provides a new idea for mangrove health monitoring by estimating the physicochemical parameters of mangroves. It provides a vital data reference for mangrove restoration and conservation in the Beibu Gulf region of Guangxi, China. Meanwhile, it provides risk warnings for mangrove ecosystems to facilitate rapid response and decision-making for mangrove conservation and restoration.

## 2. Materials and Methods

### 2.1. Study Area

The study area is located in Shajiao village (SJ) (Figure 1) along the Dafeng River in the middle zone of Beibu Gulf, Guangxi, with the geographical coordinates of $108°48'15''$ E~$108°52'15''$ E, $21°37'00''$ N~$21°38'20''$ N, which belongs to a subtropical monsoonal maritime climate. It is located at the confluence of the estuary of the Dafeng River and the inlet of the sea, where three typical mangroves are mainly distributed: *Avicennia marina* (AM), *Aegiceras corniculatum* (AC), and *Kandelia candel* (KC). The area of mangrove forest in the study area is about 1.28 km$^2$. Most of the mangrove forests in the area are natural and dominated by shrubland, and white bone loam is the dominant population in the area. In addition, the pest and disease stress and the invasion of *Sporobolus alterniflorus* in the area have led to changes in leaf traits, which are important factors causing mangrove degradation and threatening the health of the mangrove ecosystem.

### 2.2. Datasets

#### 2.2.1. Field Measurements

To obtain accurate canopy chlorophyll content values and UAV image datasets, the experiment collected 228 sample squares from 8 January 2021 to 14 January 2021 in our study area, with the following specific requirements for data collection: (1) to avoid the influence of human factors, the survey site is 30 m away from the shore; (2) the size of each sample plot was as close to a 10 m × 10 m square as possible, and existing studies have shown that a 10 m × 10 m sample plot is the best scale for estimating LCC and LAI [11,49]; (3) in the study area, we collected 85, 62, and 44 sample plots of *Avicennia marina* (AM), *Aegiceras corniculatum* (AC), and *Kandelia candel* (KC) species, respectively, each with a sample plot size of 10 m × 10 m; (4) LAI data of mangroves were measured using the LAI-2200 Plant Canopy Analyzer instrument.

The LCC data of mangrove leaves were measured with the Chlorophyll Meter SPAD-502 Plus instrument (Table 1), and the geographical locations of the collected ground points were recorded with the CNOOC V90 GNSS RTK. There was a significant correlation between SPAD values and chlorophyll content in the forest; therefore, SPAD values can effectively reflect the leaf chlorophyll content variation in mangroves [15]. The LAI-2200 Plant Canopy Analyzer is considered as an ideal tool for measuring LAI in the field [50]. The Hi-Target V90 GNSS RTK handbook differential mode has centimeter-level positioning accuracy and can provide accurate location information for each mangrove plot. The measured parameters are shown in Table 1.

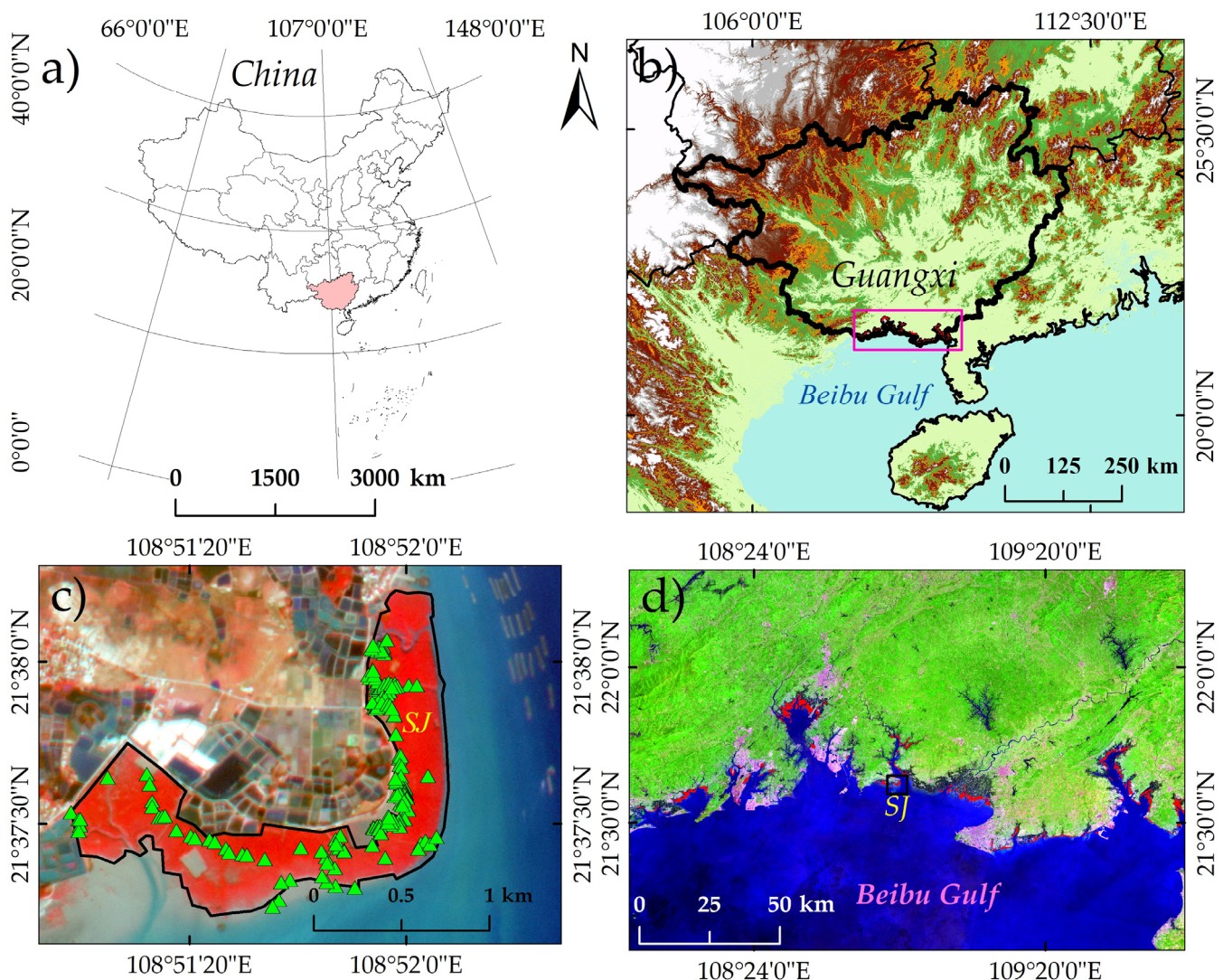

**Figure 1.** Location of study area and distribution of measured points. (**a,b**) The location of Guangxi Province, China, and the coastal zone in the study. (**c**) The map of mangrove extent in Sajiao village, a typical study area in Beibu Gulf, Guangxi, with a false color image from the combination of bands 4, 3, and 2 of GF-6; black vector linear elements mark the mangrove extent in the typical study area, and green dot elements mark the distribution of ground collected data. (**d**) The false color image of Landsat 8 OLI sensor in the Beibu Gulf area of Guangxi, with red vector border marking the mangrove area range.

**Table 1.** Statistics of LAI and LCC values derived from field measurements in the study area.

| Measured Parameters | Tree Species | Number of Samples | Mean Value | Standard Error of Mean | Numerical Range |
|---|---|---|---|---|---|
| LAI | AM | 85 | 1.87 | 0.05 | [0.93, 3.04] |
| | AC | 62 | 2.60 | 0.08 | [1.26, 4.03] |
| | KC | 44 | 1.96 | 0.07 | [0.94, 3.00] |
| | AM + AC + KC | 191 | 2.14 | 0.04 | [0.93, 4.03] |
| LCC (SPAD) | AM | 85 | 45.82 | 0.40 | [38.70, 57.00] |
| | AC | 62 | 45.71 | 0.62 | [31.90, 53.80] |
| | KC | 44 | 58.26 | 0.64 | [48.30, 71.50] |
| | AM + AC + KC | 191 | 49.93 | 0.48 | [31.90, 71.50] |

**Table 1.** *Cont.*

| Measured Parameters | Tree Species | Number of Samples | Mean Value | Standard Error of Mean | Numerical Range |
|---|---|---|---|---|---|
| LAI × LCC (SPAD) | AM | 85 | 86.24 | 2.75 | [39.50, 164.16] |
| | AC | 62 | 120.07 | 4.58 | [40.19, 192.45] |
| | KC | 44 | 114.76 | 5.15 | [52.36, 183.76] |
| | AM + AC + KC | 191 | 107.02 | 2.52 | [39.50, 192.45] |

AM + AC + KC: a collection of AM, AC, and KC species samples.

### 2.2.2. UAV and GF-6 Data Pre-Processing

To ensure the accuracy of the inversion of the mangrove canopy chlorophyll content, the UAV data were collected at the same time as the ground truth data. The DJI Matrice 200 (M200) UAV used in the experiment is equipped with the MicaSense RedEdge™ (MicaSense, Inc. Located in Seattle, WA, USA) multispectral sensor, which can provide blue (band1: 460–510 nm), green (band2: 545–575 nm), red (band3: 630–690 nm), red-edge (band4: 712–722 nm), and NIR (band5: 820–860 nm) multispectral sensors. During the operation, the flight altitude was set to 100 m, and the overlap rate of both the heading and side direction was set to 80%. Before takeoff, the radiation calibration was performed by using the standard calibration plate, and the ground absolute reflectance image with better than 0.07 m spatial resolution was finally obtained by processing with Pix4Dmapper v4.5.6 software and ENVI5.6 software (Figure 2).

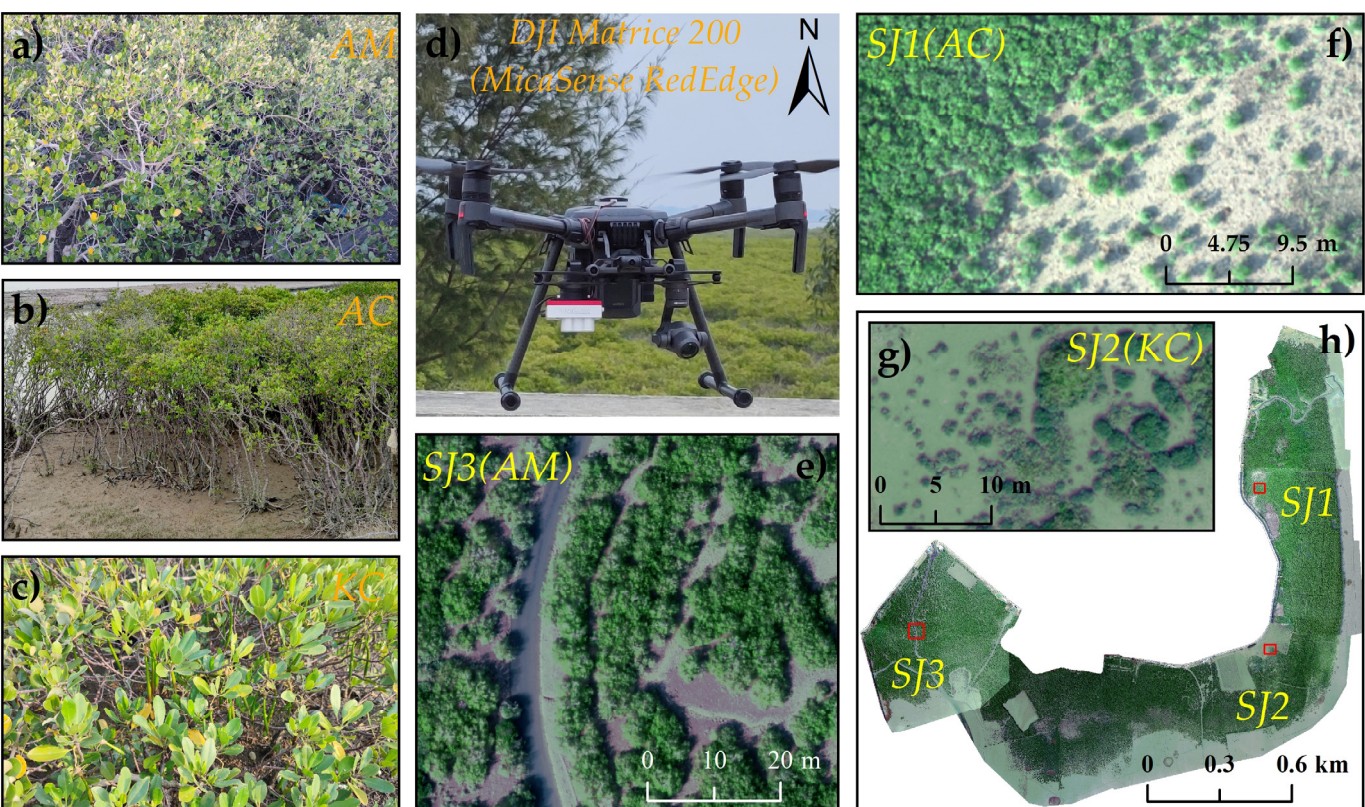

**Figure 2.** UAV images and field photos of AM, AC, and KC: (**a**), (**b**) and (**c**) indicate AM, AC, and KC photographs taken in the field; (**d**) is a photo of the UAV taking off; (**e**), (**f**) and (**g**) denote the AM, AC, and KC multispectral photographs taken by the UAV, respectively; (**h**) is a multispectral image of mangroves throughout the study area, and the red squares from left to right are shown enlarged in (**e**), (**f**), and (**g**) respectively.

The reflectance information of GF-6 satellite data is sensitive to vegetation parameters, such as LAI and LCC, and is suitable for estimating parameters related to vegetation growth to monitor vegetation health, so this paper will use GF-6 satellite data to estimate mangrove canopy chlorophyll content information [51–54]. GF-6 data were acquired through the China Resources Satellite Center (http://www.cresda.com, URL (accessed on 21 October 2022)), and the images were acquired on 2 January 2021. GF-6 data are equipped with an 8-band CMOS detector, and the satellite is equipped with 2 m panchromatic and 8 m spatial resolution multispectral high-resolution cameras, which can provide blue (band1: 450–520 nm), green (band: 520–600 nm), red (band3: 630–690 nm), NIR (band4: 760–900 nm), and Pan (p: 450–900 nm). In order to obtain accurate multispectral reflectance information with high resolution of subsatellite image elements, GF-6 data were pre-processed with radiometric calibration and atmospheric correction, and the multispectral and panchromatic images were aligned with the positioning information provided by GNSS RTK, and fused into 2 m spatial resolution using the Gram–Schmidt Pan Sharpening tool of ENVI 5.6. data containing all multispectral information.

### 2.3. Methods

RF, GBRT, and XGBoost algorithms are robust in quantitative inversion and are widely used in the estimation of mangrove parameters [15,47,55,56]. In this paper, the feature variables (reflectance bands or combined vegetation indices sensitive to CCC obtained after correlation analysis and feature variable selection) are used as input parameters for the machine learning regression algorithm, with CCC as the target variable. In this study, we attempted to invert mangrove CCC values using UAV and GF-6 data combined with random forest (RF), gradient boosting (GBRT), and extreme gradient boosting (XGBoost) algorithms to map mangrove CCC by combining UAV and MLR, GF-6, and MLR models, respectively. The overall method flow is shown in Figure 3.

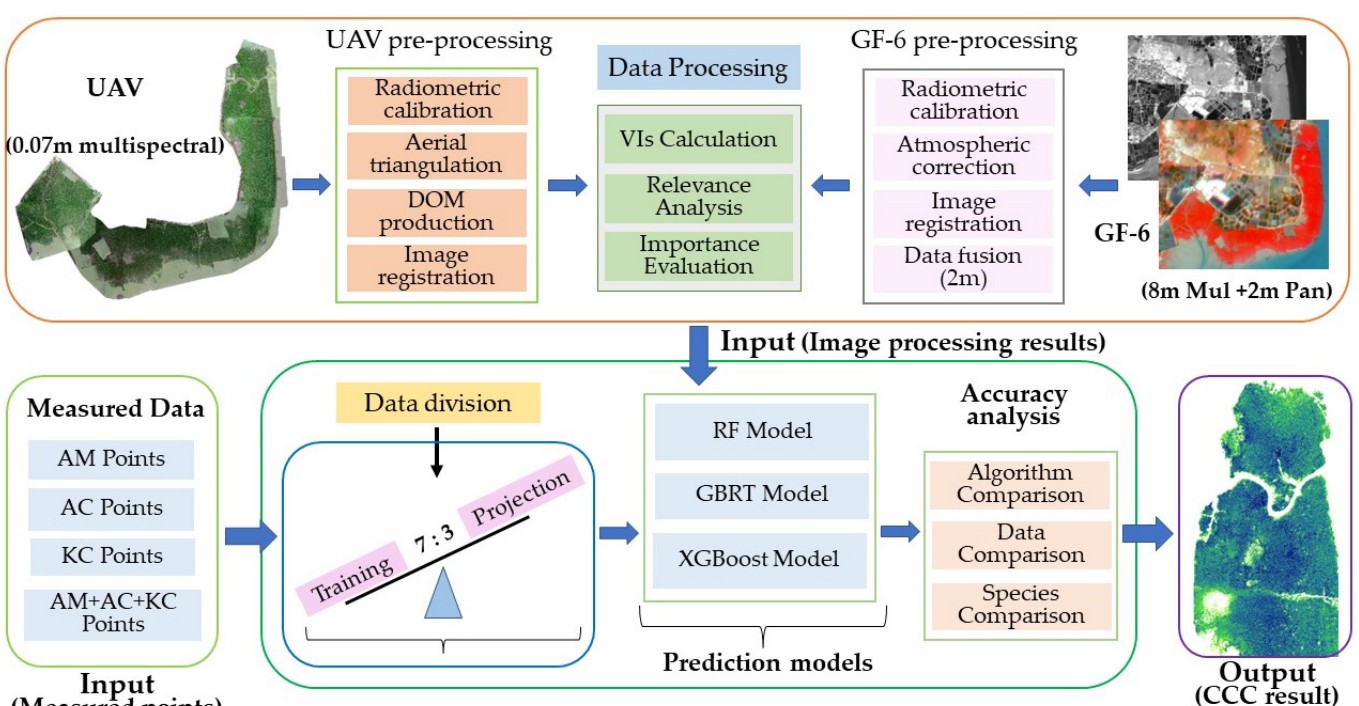

**Figure 3.** Mangrove CCC inversion flowchart.

### 2.3.1. Combined Vegetation Indices

To make full use of the reflectance information provided by UAV multispectral to establish the narrow-band combined vegetation index associated with mangrove CCC, the Band Math tool of ENVI5.6 was used to calculate the Ratio index (RI), Normalized

difference index (NDI), the ratio of single-band reflectance to the product of two bands reflectance (RSTI), and the Modified Datt index (MDATT), of which 74 and 30 combined vegetation indices were calculated for UAV and GF-6 data, respectively (Table 2). The indices were selected by the feature variables to filter the most sensitive VIs to the CCC of mangroves.

**Table 2.** Formula of combined vegetation indices.

| Vegetation Index | Formula | Reference |
|---|---|---|
| Ratio index (RI) | RI $(R_{\lambda 1}, R_{\lambda 2}) = R_{\lambda 1}/R_{\lambda 2}$ | [57] |
| Normalized difference index (NDI) | NDI $(R_{\lambda 1}, R_{\lambda 2}) = \|R_{\lambda 1} - R_{\lambda 2}\|/(R_{\lambda 1} + R_{\lambda 2})$ | [57] |
| The ratio of single-band reflectance to the product of two bands reflectance (RSTI) | RSTI $(R_{\lambda 1}, R_{\lambda 2}, R_{\lambda 3}) = R\lambda_1/(R_{\lambda 2} \times R\lambda_3)$ | [58] |
| Modified Datt index (MDATT) | MDATT $(R_{\lambda 1}, R_{\lambda 2}, R_{\lambda 3}) = (R_{\lambda 1} - R_{\lambda 2})/(R_{\lambda 1} - R_{\lambda 3})$ | [59] |

$R_{\lambda 1}$, $R_{\lambda 2}$, and $R_{\lambda 3}$: correspond to the band names corresponding to $\lambda 1$, $\lambda 2$, and $\lambda 3$ (450–900 nm) wavelengths of the multispectral data, respectively.

### 2.3.2. Machine Learning Regression Algorithms

Random Forest (RF) is a very representative Bagging integration algorithm, where all its underlying evaluators are decision trees, and its main purpose is to make predictions on new data points by performing voting [60]. A decision tree is a tree-like structure in which each internal (non-leaf) node is labeled with a test for some attribute; for each possible test result, there is an arc leading to a unique (child) node, and each leaf node of the tree is labeled with the value to be predicted [61]. For a forest of regression trees, the resulting predictions are usually formed by averaging the individual predicted values [62,63].

The gradient boosting (GBRT) algorithm is a robust regression algorithm that integrates many regressors, and its main idea is to build a new regression tree in the direction of decreasing the gradient of the loss function based on the results of the previous regression tree [64]. The key approach of angle boosting is to increase the approximation of the residuals in the tree algorithm with the help of negative gradient values of the loss function and then fit the regression tree [65]. The GBRT model has no restrictions on any input data assumptions and has better predictive performance and stability than a single decision tree by integrating the algorithm [66].

The extreme gradient boosting (XGBoost) algorithm is a scalable tree-boosting method by creating several decision trees that are made using the prediction errors and residuals of previous tree models, rather than averaging over independent trees [67]. However, compared to simple gradient boosting algorithms, the XGBoost algorithm differs in that the process of adding weak learners is not performed individually and takes a multi-threaded approach that makes proper use of the machine's CPU cores, resulting in faster speeds and better performance [68].

### 2.3.3. Improvement of Model Parameters

The optimal parameters of the machine learning regression models were determined using the grid search method. The model tuning was performed in the Tensorflow 2.0 environment using the GridSearchCV function of the Python 3.7.10 software, and the optimal parameters for each model were finally traversed as follows: the number of decision trees (n_estimators) for the RF model was 250, the minimum number of samples (min_samples_split = 4) required for the internal nodes in the division n_estimators = 150, max_depth = 1, min_samples_split = 5 for GBRT model, n_estimators = 100, max_depth = 1 for XGBoost model, and n_estimators = 100, max_depth = 1 for XGBoost model.

### 2.3.4. Accuracy Metrics

For the accuracy assessment of the inversion process, the study data were divided into a training set and a test set in the ratio of 7:3. The training set is used to train and evaluate the overlaid machine learning regression model, and the test set is mainly used as

validation data to evaluate the accuracy of the prediction maps. In this paper, the coefficient of determination ($R^2$) is used to test how close the dataset is to the fitted regression line and to evaluate the predictive performance of the regression model. The root-mean-square error (RMSE) tests the correlation between predicted and measured chlorophyll values.

In this paper, the coefficient of determination ($R^2$) can reflect the degree of fit between the estimated values of the RF, GBRT, and XGBoost models and the ground truth through the regression relationship, to judge the ability of each machine learning regression model to invert the mangrove CCC. The expression of $R^2$ is shown in Equation (1).

$$R^2 = 1 - \frac{SSE}{SST} = 1 - \frac{\sum_{i=1}^{n} (\hat{y}_i - y_i)^2}{\sum_{i=1}^{n} (\overline{y} - y_i)^2} \tag{1}$$

where $R^2$ is the coefficient of determination of the measured and predicted values in estimating the mangrove canopy chlorophyll content, $SSE$ is the sum of the squared residuals of the predicted mangrove canopy chlorophyll content and each of the input measured values, $SST$ is the sum of the squared deviations of the mean of the input measured values of the mangrove canopy chlorophyll content and each of the input measured values, $\hat{y}_i$ is the input measured data of the canopy chlorophyll content, $y_i$ is the measured value of each input, and $\overline{y}$ is the mean value of each input measured value.

The estimated value of CCC is compared to the actual measured value using the root-mean-square error (RMSE), which is a measure of the deviation between the predicted value of CCC (obtained by inversion of a machine learning regression model) and the actual ground measurements [69]. The RMSE can characterize the degree of curve fit between the predicted and measured values, and is used to measure the accuracy of the model prediction; the smaller the RMSE value, the higher the accuracy of the prediction. The expression of RMSE is shown in Equation (2).

$$RMSE = \sqrt{\frac{\sum_{i=1}^{n} (\hat{y}_i - y_i)^2}{n}} \tag{2}$$

where $RMSE$ is the root-mean-square error between the predicted value and the ground truth values when inverting the mangrove canopy chlorophyll content, $n$ is the total number of data input test data or training data when estimating the mangrove canopy chlorophyll content, $\hat{y}_i$ is the predicted values of the mangrove canopy chlorophyll content test data or training data, and $y_i$ is each test data or training data input.

## 3. Results

### 3.1. Correlation Analysis to Obtain the Sensitivity of Vegetation Indices to CCC

To determine the correlation between the feature variables extracted from each image and the CCC of mangroves, the top 12 correlation indices of each mangrove species from UAV and GF-6 data were intercepted and plotted in the correlation heat map (Figures 4 and 5) and the semicircle pie chart (Appendix A).

The feature indices calculated from UAV airborne multispectral images had a higher correlation with the CCC of each type of mangrove species, and the sensitivity was generally higher than that of GF-6 satellite data. The correlation coefficients of RI35 and NDI35 with the CCC of AM were the highest, with a correlation coefficient of 0.803; the correlation coefficients of RSTI413 and MDATT413 with the CCC of AC were both 0.891, with the highest sensitivity; the correlation coefficients of NDI35 and RI35 with the CCC of KC were the highest, with a correlation coefficient of 0.7. The highest correlation coefficients were 0.766. RSTI324 and RSTI412 feature indices of GF-6 data had the highest correlation with the CCC of AM and AC, with correlation coefficients of −0.774 and 0.798, respectively. The

feature index with the highest correlation between GF-6 and the CCC of KC was RSTI214, with a correlation coefficient of only −0.461, which had a low sensitivity.

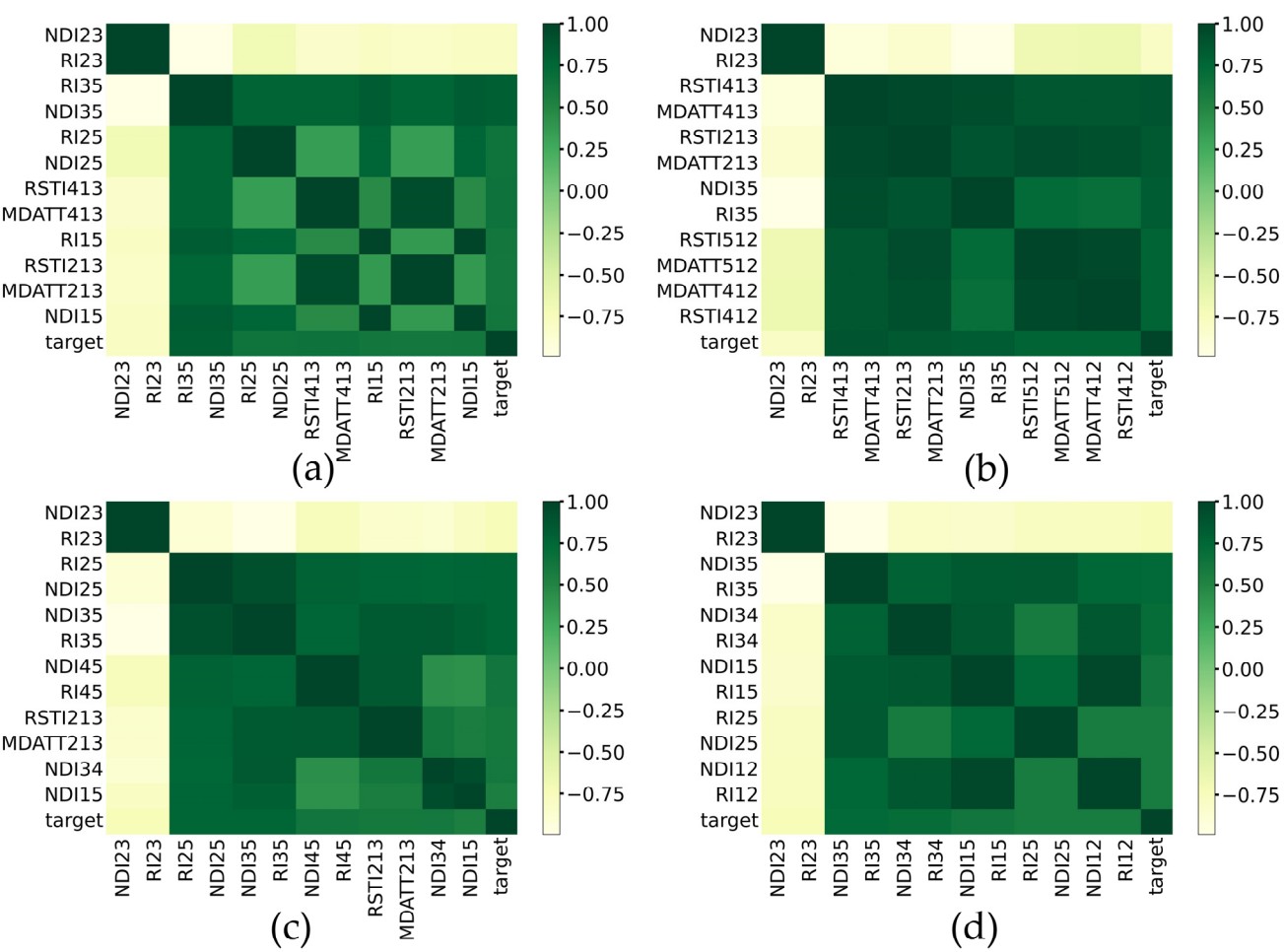

**Figure 4.** Heat map of correlation among AM, AC, KC, and AM + AC + KC characteristic indices in UAV image. (**a**) Heat map of correlation of AM specie; (**b**) Heat map of correlation of AC specie; (**c**) Heat map of correlation of KC specie; (**d**) Heat map of correlation of AM + AC + KC species. The numbers "23", "25", and "213" mean the name of the band corresponding to the combined vegetation index, such as NDI23, which represents the combined vegetation index consisting of band 2 and band 3 in UAV data.

Among the inversions of all mangrove species, the NDI23 and RI23 characteristic indices of UAV data had the highest correlation coefficients of −0.737 for CCC of mangroves, while the optimal feature variables in GF-6 data were RSTI312, NDI23, and RI34, with correlation coefficients of −0.488, −0.484, and 0.480, respectively, with correlation coefficients less than 0.5, with low sensitivity.

### 3.2. Feature Importance Assessment to Obtain Optimal Feature Variables

To select the best parameters to participate in model training, the importance ranking of the feature variables in the UAV and GF-6 data needs to be obtained. The Scikit-learn machine learning library for Python provides feature selection methods that can be used to output feature importance scores. This is done through the SelectFromModel class, which uses a (machine learning regression) model that transforms the dataset into a subset with selection elements. In this paper, the XGBoost model trained on the training set was selected for feature variable selection, and the feature variables that contributed most to the CCC inversion results of mangrove AM, AC, KC, and AM + AC + KC species in the

UAV and GF-6 data were selected by evaluating the importance scores, and the importance scores of each feature variable are shown in Figure 6, and Appendix B.

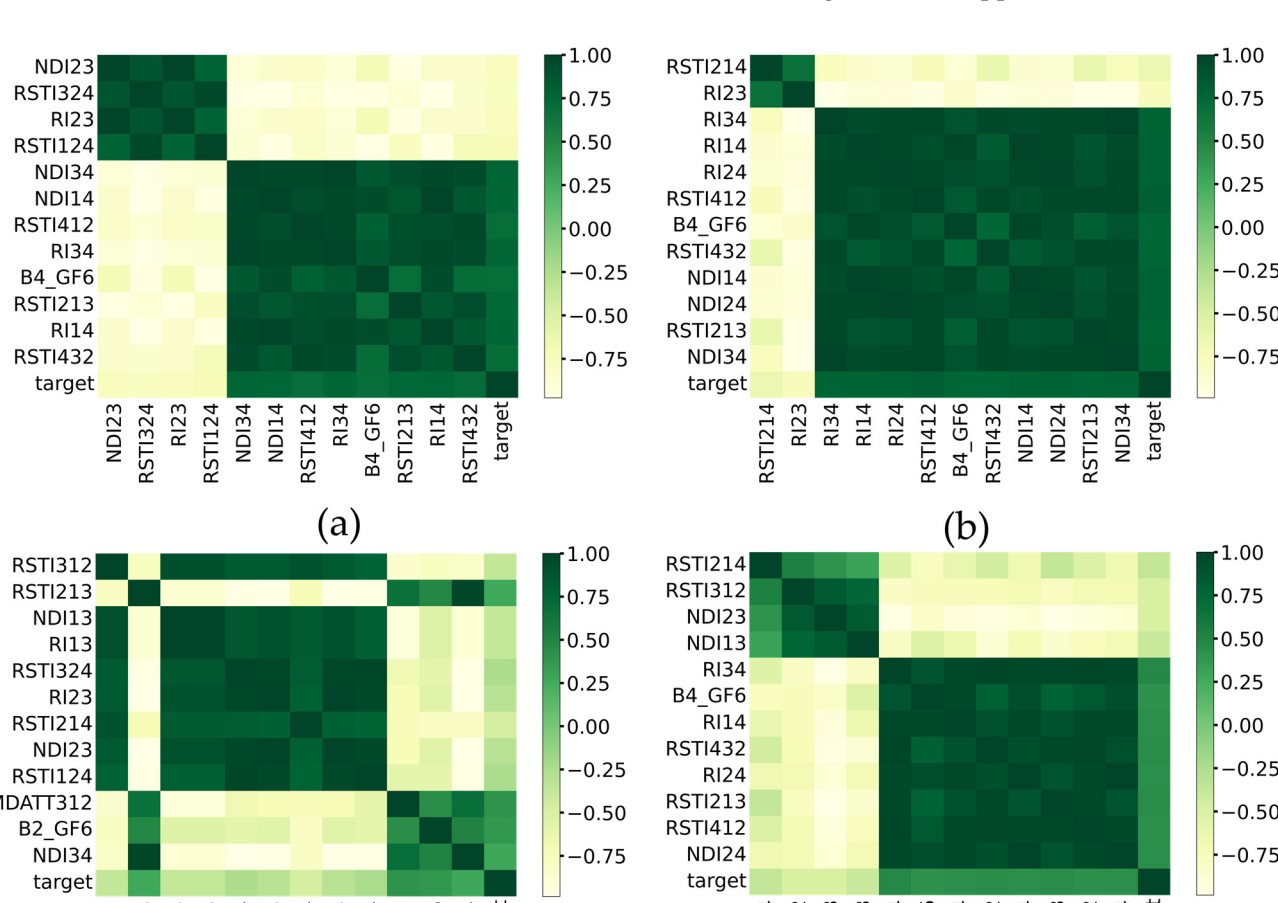

(a)

(b)

(c)

(d)

**Figure 5.** Heat map of correlation among AM, AC, KC, and AM + AC + KC characteristic indices in GF-6 image. (**a**) Heat map of correlation of AM specie; (**b**) Heat map of correlation of AC specie; (**c**) Heat map of correlation of KC specie; (**d**) Heat map of correlation of AM + AC + KC species.

(1) In the UAV data. The feature variables RI35, RI23, and NDI35 contributed the most to the inversion of CCC of AM mangrove species, with importance scores of 8.509, 8.391, and 8.108, respectively; MDATT413, RSTI413, and RSTI213 had the highest importance scores in the inversion of CCC of AC species with 9.349, 9.032, and 8.045, respectively; RI35, NDI35, and NDI25 had importance scores of 6.920, 6.780, and 6.233, respectively. RI35, NDI35, and NDI25 with importance scores of 6.920, 6.780, and 6.233, respectively, were the best feature variables for inversion of CCC of KC mangrove species. The most sensitive characteristic indices for canopy chlorophyll content among AM + AC + KC species were NDI35, RI35, and RI23, with characteristic importance scores of 10.035, 9.950, and 9.923, respectively.

(2) In the GF-6 images. The feature variables with the highest contribution to the inversion of the CCC of AM mangrove species were RSTI324, RI34, and NDI34, with importance scores of 9.419, 8.988, and 8.846, respectively; the importance scores of the B4 NIR band, NDI14, and RI14 feature indices were 7.512, 7.341, and 7.322, respectively, which were the best feature variables for the inversion of the CCC of AC. The importance scores of RSTI124, RSTI214, and RSTI324 were only 2.420, 2.355, and 2.340, respectively. The importance scores of NDI23, RI34, and RSTI312 in AM + AC + KC species were 10.827,

8.149, and 7.289, respectively, which were the optimal feature variables for the inversion of AM + AC + KC.

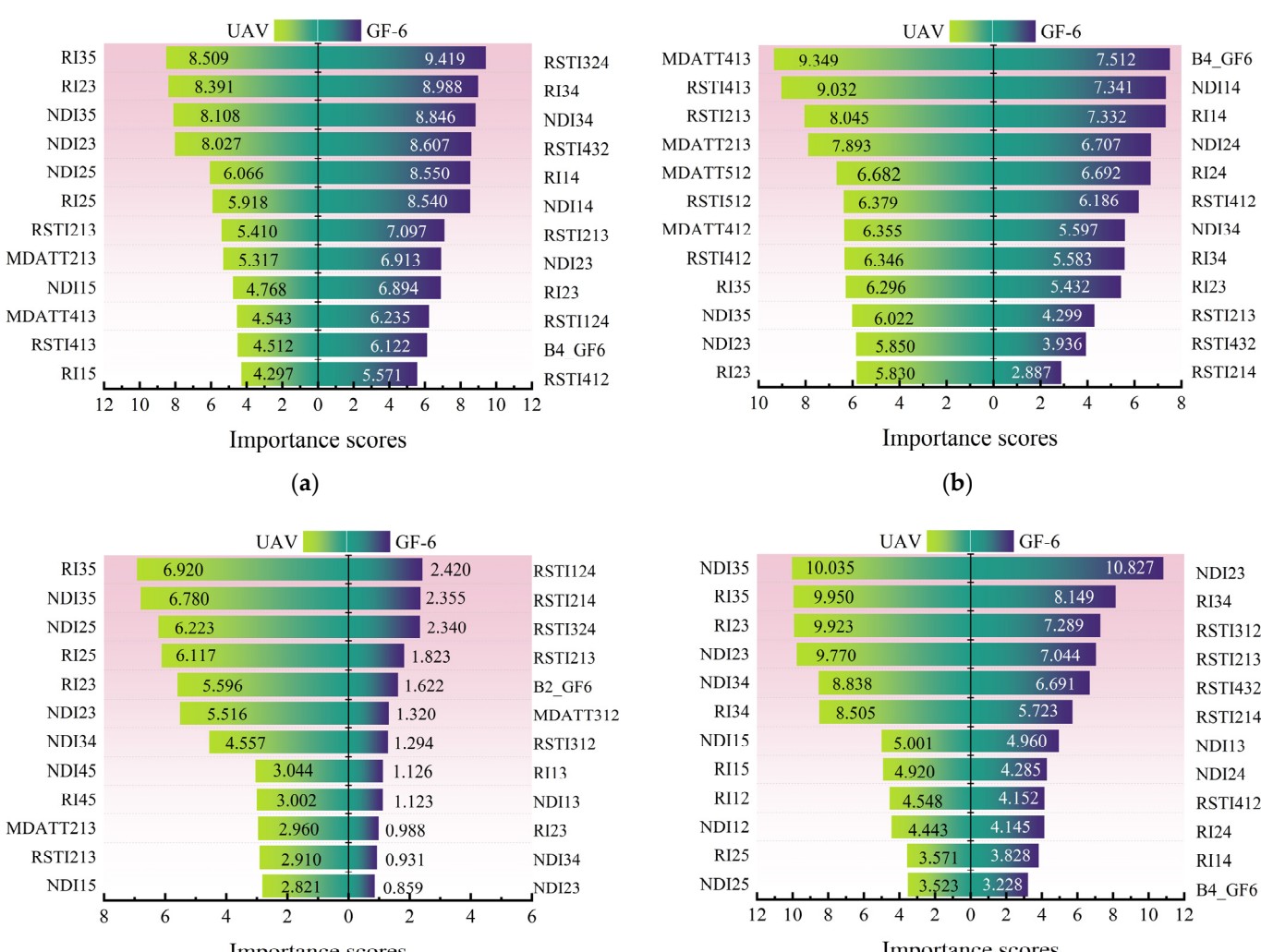

**Figure 6.** Stacked bar charts of the importance scores of feature indices in UAV, and GF-6 data. (**a**) Importance scores of UAV and GF-6 data in AM; (**b**) Importance scores of UAV and GF-6 data in AC; (**c**) Importance scores of UAV and GF-6 data in KC; (**d**) Importance scores of UAV and GF-6 data in AM + AC + KC.

### 3.3. Accuracy Analysis to Obtain Optimal Model Algorithms and Data for Estimation

In this paper, the training and validation sets were divided into the ratio of 7:3, and the validation samples of AM, AC, and KC mangrove species CCC were predicted by the RF, GBRT, and XGBoost models, respectively, and the fitness of each machine learning model for mangrove species was judged by the degree of the curve fit and accuracy assessment metrics, such as $R^2$ and RMSE (Figures 7 and 8).

(1) AM species. There is an optimal fit between the estimated and predicted values of RF and GBRT. The $R^2$, RMSE, and r estimated by the RF regression model of UAV are 0.764, 17.762 SPAD, and 0.874, respectively, and the inversion accuracy of the GBRT regression model of GF-6 is $R^2 = 0.624$, RMSE = 21.498 SPAD, r = 0.790; at points 4 and 25 of the fitting curve between the predicted and measured values of UAV data, the fitting effect of the RF regressor is significantly better than that of the GBRT and XGBoost algorithms, and from the accuracy assessment index, the $R^2$ of the RF regressor compared to GBRT and XGBoost is improved by 0.158 and 0.120, and the RMSE was reduced by 4.697 and 2.987

SPAD, respectively. In the GF-6 data, the difference in the fitting effect of the estimated values of GBRT and the RF regressor to the measured results was smaller, and the fitting effect of the GBRT algorithm was better than the XGBoost algorithm at points 3 and 23, and the $R^2$ was improved by 0.064 and the RMSE was reduced by 2.403 SPAD; the fitting effect of the RF model curve in UAV was better than that of the GBRT model for GF-6 data at the point numbers 16 to 23.

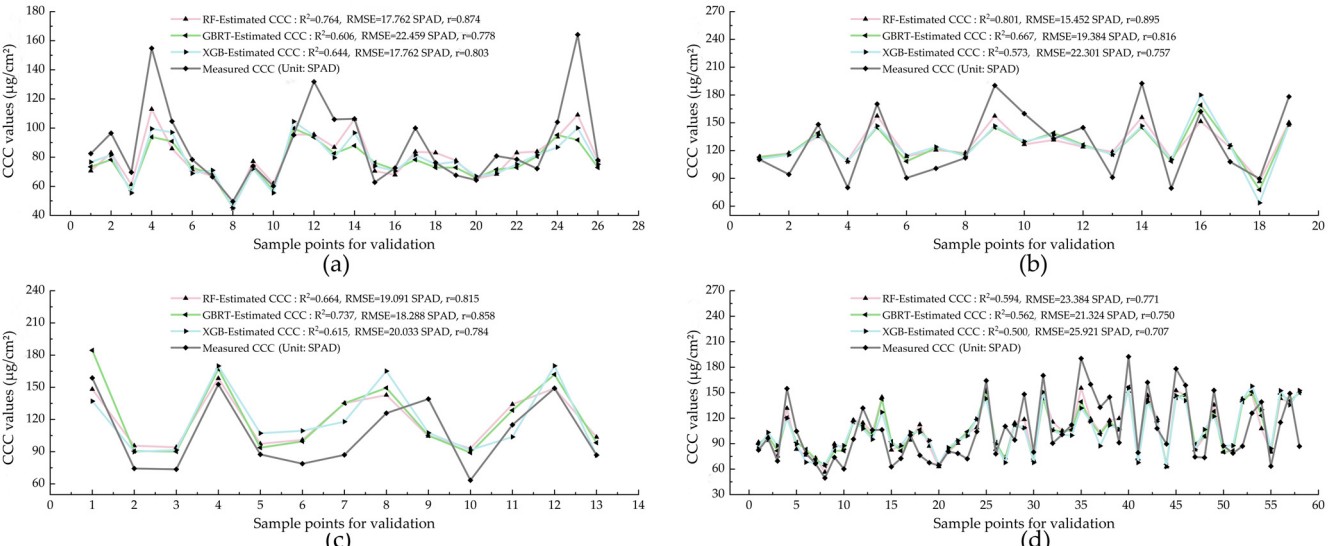

**Figure 7.** Accuracy of fitting curves for inversion of MLR in UAV data. (**a**) Fitting accuracy of AM; (**b**) Fitting accuracy of AC; (**c**) Fitting accuracy of KC; (**d**) Fitting accuracy of AM + AC + KC.

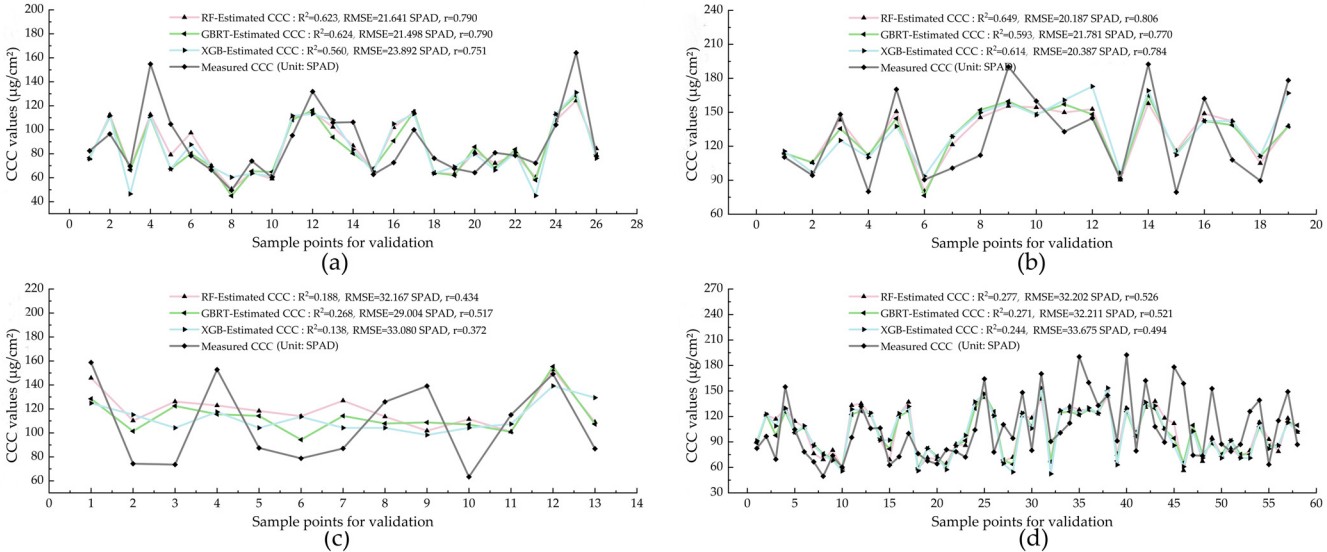

**Figure 8.** Accuracy of fitting curves for inversion of MLR in GF-6 data. (**a**) Fitting accuracy of AM; (**b**) Fitting accuracy of AC; (**c**) Fitting accuracy of KC; (**d**) Fitting accuracy of AM + AC + KC.

(2) AC species. The RF model in UAV and GF-6 data had the highest fitting accuracy with $R^2$ and RMSE of 0.801, 15.452 SPAD and 0.649, 20.187 SPAD, respectively; in UAV data, the fitting accuracy of the RF regression model was generally better than that of GBRT and XGBoost, with $R^2$ improved by 0.134, 0.228, and RMSE improved by 3.932, 6.8493.718 SPAD, respectively. In the GF-6 data, the overall effect of the RF fitted curve was better than GBRT and XGBoost, and in the accuracy index, the $R^2$ and RMSE of the RF regressor improved by 0.056, 1.594 SPAD; compared to the XGBoost model, $R^2$ improved by 0.035

and RMSE decreased by 0.200 SPAD. In the fitted curves of the RF regression model for UAV and GF-6 data, the fitting effect of UAV data was significantly better than that of GF-6 data for the 6 to 12 point positions.

(3) KC species. The GBRT model fitted best in the UAV data, with $R^2$ of 0.737 and RMSE of 18.282 SPAD, while the GF-3 data had very low correlation between the feature variables and CCC, resulting in poor fitting of each model, which could not effectively invert the CCC of mangrove KC species. The fitting effect of the fitted curve of the GBRT model was better than that of the RF and XGBoost models in general, with an increase of 0.073 in $R^2$ and a decrease of 0.8031 SPAD in RMSE compared to the RF model, and an increase of 0.122 and 1.745 SPAD in $R^2$ and RMSE accuracy compared to the XGBoost model, respectively.

(4) AM + AC + KC species. The RF model has a better fit in the inversion. Among the prediction fitting curves of each model for UAV data, RF had the best overall fitting accuracy, which was significantly better than the GBRT and XGBoost models at point 5. The RMSE of the RF model was 0.940 SPAD lower than GBRT and 2.537 SPAD lower than the XGBoost model. In GF-6 data, due to the feature variables having low species correlation for AM + AC + KC, resulting in the lower fitting accuracy of each model prediction to the measured data, the $R^2$ of the optimal model RF was 0.277 and RMSE was 32.202 SPAD.

The CCC results of the inversion of the optimal machine learning regression model selected among mangrove species were compared with the measured data by one-dimensional linear regression analysis to compare the inversion accuracy among multiple sources of data and mangrove species, as shown in Figure 9.

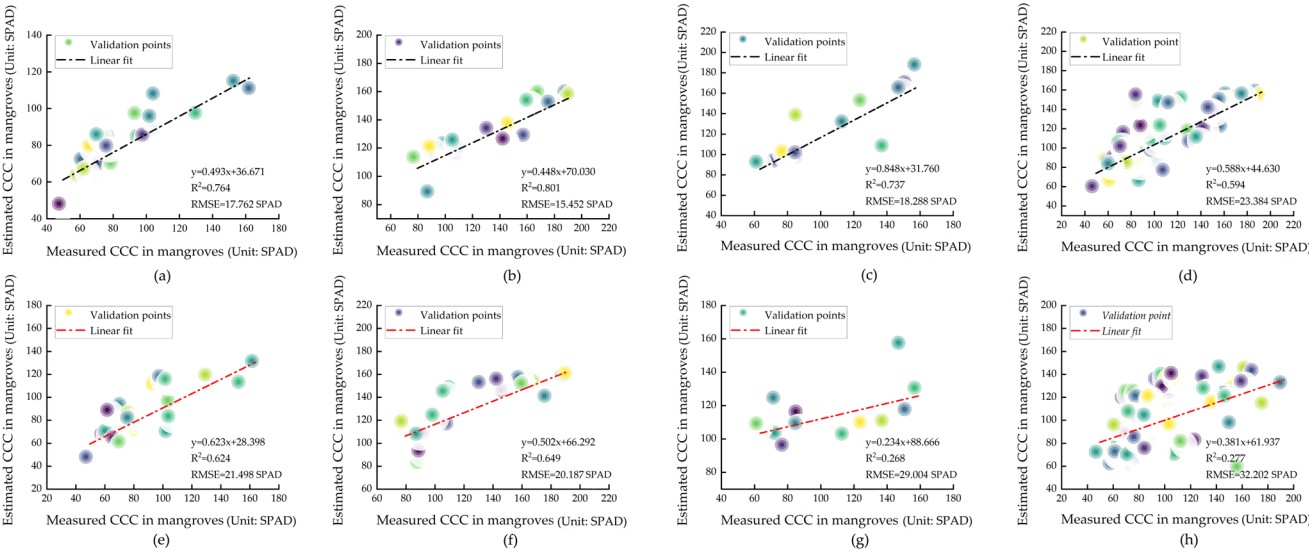

**Figure 9.** Accuracy of optimal machine learning regression models for inversion of mangrove CCC in UAV and GF-6 data. (**a**) Prediction accuracy of UAV data feature variables in AM; (**b**) Prediction accuracy of UAV data feature variables in AC; (**c**) Prediction accuracy of UAV data feature variables in KC; (**d**) Prediction accuracy of UAV data feature variables in AM + AC + KC; (**e**) Prediction accuracy of GF-6 data feature variables in AM; (**f**) Prediction accuracy of GF-6 data feature variables in AC; (**g**) Prediction accuracy of GF-6 data feature variables in KC; (**h**) Prediction accuracy of GF-6 data feature variables in AM + AC + KC.

In the accuracy comparison between UAV and GF-6 data, the accuracy of UAV data was significantly better than the inversion accuracy of GF-6 data. In the inversion of CCC for AM species, the $R^2$ of UAV data combined with the optimal model improved by 0.140, and the RMSE decreased by 3.736 SPAD than that of GF-6 data; in the inversion of CCC for AC species, the $R^2$ of UAV data improved by 0.125, and the RMSE decreased by 4.735 SPAD than that of GF-6 data; for KC species, the $R^2$ was 0.737, and RMSE was 18.288 SPAD,

while the GF-6 data's could not effectively invert the mangrove CCC, with $R^2 < 0.3$ and RMSE > 29 SPAD. Based on the inversion results for all mangrove species, the UAV data still achieved high accuracy, with $R^2 = 0.594$ and RMSE = 23.384 SPAD, while the GF-6 data had the lowest precision, with RMSE > 32 SPAD.

In the accuracy comparison among AM, AC, and KC species, AC species achieved the best results, with $R^2 > 0.80$ and 0.64 in the inversion results of UAV and GF-6 data, respectively; KC species had the lowest accuracy of CCC estimation, with RMSE > 29 SPAD for GF-6 data, which could not effectively invert the mangrove CCC, and after integrating all species, the accuracy of inversion was lower than that of single mangrove species. Based on the estimation results of AM + AC + KC optimal machine learning in the UAV data, the mangrove CCC inversion equation was obtained as follows.

$$CCC_{(AM+AC+KC)} = 0.588x + 44.630 \tag{3}$$

where $x$ is the set of predicted values output by the joint UAV data and the optimal machine learning regression model.

### 3.4. Mapping of Mangrove Canopy Chlorophyll Content

The mangrove UAV CCC habitats in a typical study area SJ were mapped according to the relationship of Equation (3) (Figure 10). As the color deepens (yellow color transitions to dark blue), the value of CCC becomes larger. The overall distribution of CCC values in the study area shows lower CCC values in the nearshore, offshore, and near-river areas, and the transition from low–high–low CCC values from the nearshore to offshore direction.

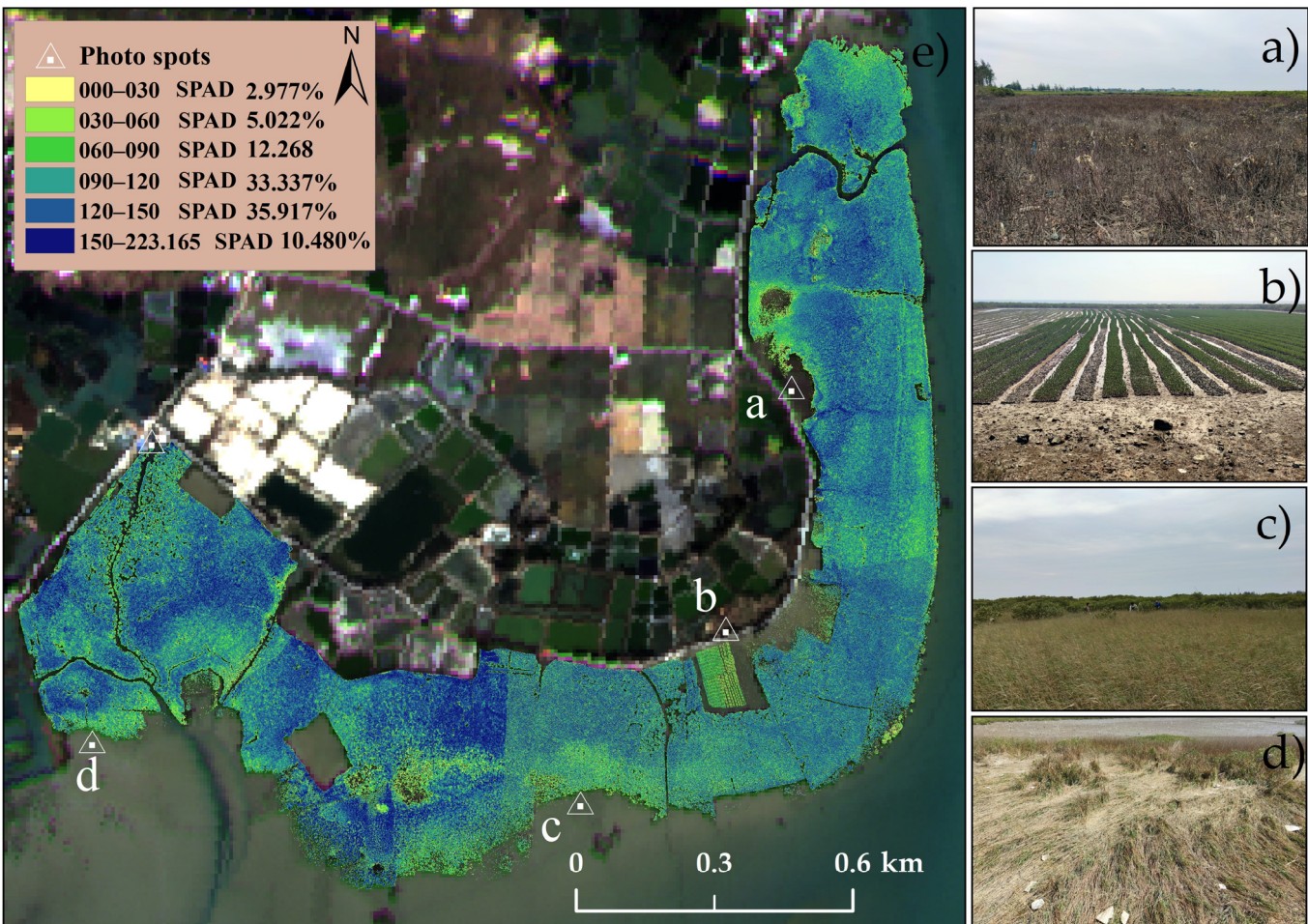

**Figure 10.** (**a**,**b**,**c**,**d**) are the photo acquisition points at (**e**); (**e**) Spatial distribution of mangrove CCC.

The CCC values of mangroves in the study area ranged from 0 to 223.165 SPAD, mainly distributed in the range of 120–150 SPAD, which accounted for 35.9% in the whole study area. Overall, the CCC was high throughout the study area, but local areas still faced mangrove health problems: (1) Large traces of vines were found at the dead mangrove in Figure 10a, and animals, such as goats, were occasionally found to gnaw at the mangrove in the area; (2) Figure 10b belonged to artificially planted seedlings, and the local management took measures to restore the area where the mangrove was damaged with artificially planted seedlings; (3) Figure 10c,d were under the stress of *Sporobolus alterniflorus*, and the vigorous *Sporobolus alterniflorus* occupied the living space of the mangrove outside the tidal flats. The CCC values of mangroves in these areas are low, so the magnitude of CCC values can reflect the health of mangroves, while the likelihood of their habitats being at risk is mapped when the CCC values are low.

## 4. Discussions

### 4.1. Spectral Characteristics and Vegetation Indices Sensitivity of Mangrove CCC

The chlorophyll content is one of the important factors affecting the intensity and rate of photosynthesis, and different spectral ranges have a large impact on mangrove physiology [70]. For example, in the blue band, the chlorophyll and carotenoid absorption ratio is the largest and has the greatest effect on photosynthesis, while in the red band, chlorophyll absorption is low and has a significant effect on photosynthesis and photoperiodic effects. Mangroves have different characteristics in the spectra of different species, and these optical feature differences can be used for quantitative inversion among species. From the UAV orthorectified and GF-6 remote sensing images, the mangrove images and spectral features between different species are obvious, among which the differences in features between different species in the UAV images are especially obvious. The GF-6 images cannot show detailed information of interspecies differences well due to the low resolution, but it can be found to have the ability to separate AM, AC, and KC through the spectral curve (Table 3), thus enabling quantitative inversion at the species-scale. In the canopy spectra of different types of healthy mangrove forests, the spectral curves generated by different optical sensors conform to the typical vegetation spectral reflectance characteristics, and there are obvious differences between AM, AC, and KC species in UAV and GF-6 images at the canopy level. The differences between mangrove species in the near-infrared band are more obvious.

The visible band of the vegetation reflectance spectrum is characterized by strong absorption of red and blue light by chlorophyll a and weak reflection of green light, while in the near-infrared band, the reflectance increases sharply due to multiple reflections of the leaves, which is due to the cessation of chlorophyll absorption above 700 nm and increased scattering by the leaf structure, while within the red-edge band (680–750 nm), there is a clear chlorophyll a absorption valley, forming a typical spectral characteristic curve [71,72]. Combined with the experimental field measurements, the mean magnitude of CCC content for each mangrove species was $CCC_{AC} > CCC_{KC} > CCC_{AM}$, and because the canopy of AC species has a higher chlorophyll content, it exhibits stronger reflectance in the near-infrared and green-band spectral ranges. In the spectral curves of KC species, a significant difference was observed in the spectral curves of GF-6 when compared with UAV data. Soil influence is prevalent in partially vegetated canopies, where soils are known to have lower reflectance in the NIR band, and GF-6 has lower spatial resolution than UAV data, which increases the possibility that individual image elements contain environmental noise, and the presence of soil factors in GF-6 data causes the spectral profiles of AC species to exhibit significant differences that largely affect the inversion results [73–75]. The reason that the GF-6 data could not effectively invert the CCC of KC species also lies in the fact that the plants of KC species in the study area are short and sparsely distributed (Figure 2h), and emphasis should be placed on avoiding these areas if the spatial resolution of the images used is low during the field experiment.

**Table 3.** Interpreted signatures and spectral characteristics of AM, AC, and KC.

| Sensor Types | Measured Points | Image Characteristics (RGB) | Interpretation Features | Species Type | Spectral Curves |
|---|---|---|---|---|---|
| UAV MicaSense RedEdge™ (0.07 m) | 85 | | Plants are tall, irregularly shaped, greenish-white in color | AM | |
| | 62 | | Plants are firm, flat, yellow-green | AC | |
| | 44 | | Plants are sparse, regular, dark green | KC | |
| GF-6 PMS (8 m multi-spectral and 2 m panchromatic fusion) | 85 | | Plants are irregular, ruffled | AM | |
| | 62 | | Plants are firm, dark green | AC | |
| | 44 | | Plants with gaps and golden green color | KC | |

When estimating vegetation parameters, the red-edge and near-infrared bands are critical when considering the most sensitive bands. It has been shown that the NIR band is more sensitive to the chlorophyll content of vegetation, while the 850 nm reflectance as the NIR reference band is more sensitive to the chlorophyll content than the 750 nm reflectance [44,72]. According to the experimental results, the red-edge and near-infrared band indices consisting of the red-edge band and near-infrared, etc. contributed more to the CCC inversion results for each mangrove species. The UAV data, RI35, MDATT413, and NDI35 bands consisting of the near-infrared (band5) and red-edge band (band4) were the most sensitive to mangrove CCC. Compared with the UAV data, the GF-6 data only have vegetation indices composed of visible and near-infrared bands, and the red-edge band, which is sensitive to vegetation chlorophyll, is missing, and the sensitivity to mangrove CCC is lower than that of the UAV data.

### 4.2. Optimal Inversion Model for Mangrove CCC

In the results of the mangrove optimal model evaluation, all three machine learning regression models effectively estimated the mangrove CCC, and as seen in Figures 8 and 9, although the values predicted by the four models have relatively high consistency, the stability and robustness of the RF and GBRT models are significantly better than the XGBoost algorithm in terms of predictive power performance. The RF and GBRT algorithms are widely used because of their advantages [76–79]. Unlike the standard regression tree, where each node is created using the best partition between all variables, RF has a randomly selected subset of variables at the nodes, with the specific size of the subset being the parameter Mtry [77]. Although this approach seems contradictory, it has the relatively best performance compared to GBRT and XGBoost in combining UAV data to invert mangrove AM, AC, and AM + AC + KC species, and GF-6 data to invert mangrove AC species for accuracy estimation. In the comparison of model accuracy for predicting mangrove AM species using UAV, and GF-6 data, the GBRT algorithm has the best prediction performance, which is because the GBRT model is more suitable for the prediction of small datasets [80]. Although the XGBoost algorithm has been further

enhanced from GBRT with a second-order Taylor spread for the loss function, inclusion of regular terms in the objective function, support for parallelism, and automatic processing of missing values, it is prone to overfitting problems, meaning that the model is too accurate and can effectively predict existing data, but cannot reliably predict future data [66]. In the inversion of CCC of mangrove species with few samples, the RF and GBRT algorithms are effective in avoiding overfitting to some extent.

For the inversion of some mangrove species, we found that the accuracy of the RF model was better than that of the GBRT and XGBoost models, e.g., Figure 7a. Through Figure 11, we learned how RF (Figure 11a), GBRT (Figure 11b), and XGBoost (Figure 11c) were fitted to the training data during the model training. For the training data, the matched points of both GBRT and XGBoost can be connected to a line. Although the correlation coefficient reaches 1, the results in the final test are contrary to the training results, and there is an overfitting situation. The RF model, on the other hand, made predictions based on the distribution of each data point as much as possible according to the distribution of the training data points, without overfitting, and achieved a high prediction accuracy.

Although RF has good local performance for the estimation of CCC in mangroves, in the complex environment of mangroves, there is a greater need for regression models that are suitable to fit all types of training sample sizes and generalize well across mangrove species, and model fusion and stacking algorithms may be one of the effective means to improve the inversion uncertainty of machine learning regression models.

Artificial intelligence technology is one of the current hot spots in the development of science and technology. In the inversion of mangrove parameters using machine learning regression algorithms, studying formal approaches for AI-based technique verification will be an important direction for future development [81,82].

*4.3. Mangrove Risk Prevention and Protection Measures*

Since 2001, the area of mangroves has increased by 1.8% per year due to strict protection and large-scale restoration of the remaining mangroves, and by 2019, 67% of China's mangroves had been enclosed within protected areas, but 33% of the area remains outside protected areas, and the remaining mangroves are suffering from extensive degradation due to widespread anthropogenic disturbances [83]. Mangroves are under stress from growing coastal populations from the land edge and rising sea levels from the ocean edge, and these pressures include the synergistic effects of seawall construction, aquaculture, overfishing, sea level rise, extreme climatic events, ecological invasions, and pollution; all of these drivers interact to potentially lead to mangrove degradation and potential future loss [83–85].

Changes in canopy chlorophyll content as an indicator of mangrove health are closely related to mangrove health risk factors. For example, the chlorophyll content of the mangrove canopy in Figure 10a–d was significantly reduced relative to other areas. The main reason for the decrease in canopy chlorophyll content in Figure 10c,d is the growth of *Sporobolus alterniflorus*, and the continued expansion of *Sporobolus alterniflorus* needs to be guarded against. In Figure 10a, the traces of human activities are more obvious, and the mangrove forest shows extensive degradation, which should be adopted as an artificial afforestation policy. At the same time, the relevant departments can protect and restore mangroves in a targeted manner according to the areas where the chlorophyll content of the mangrove canopy has decreased. The healthy growing seedlings in Figure 10b, which are in the post-afforestation period, are relatively weak against disturbing external factors and must be continuously monitored and protected.

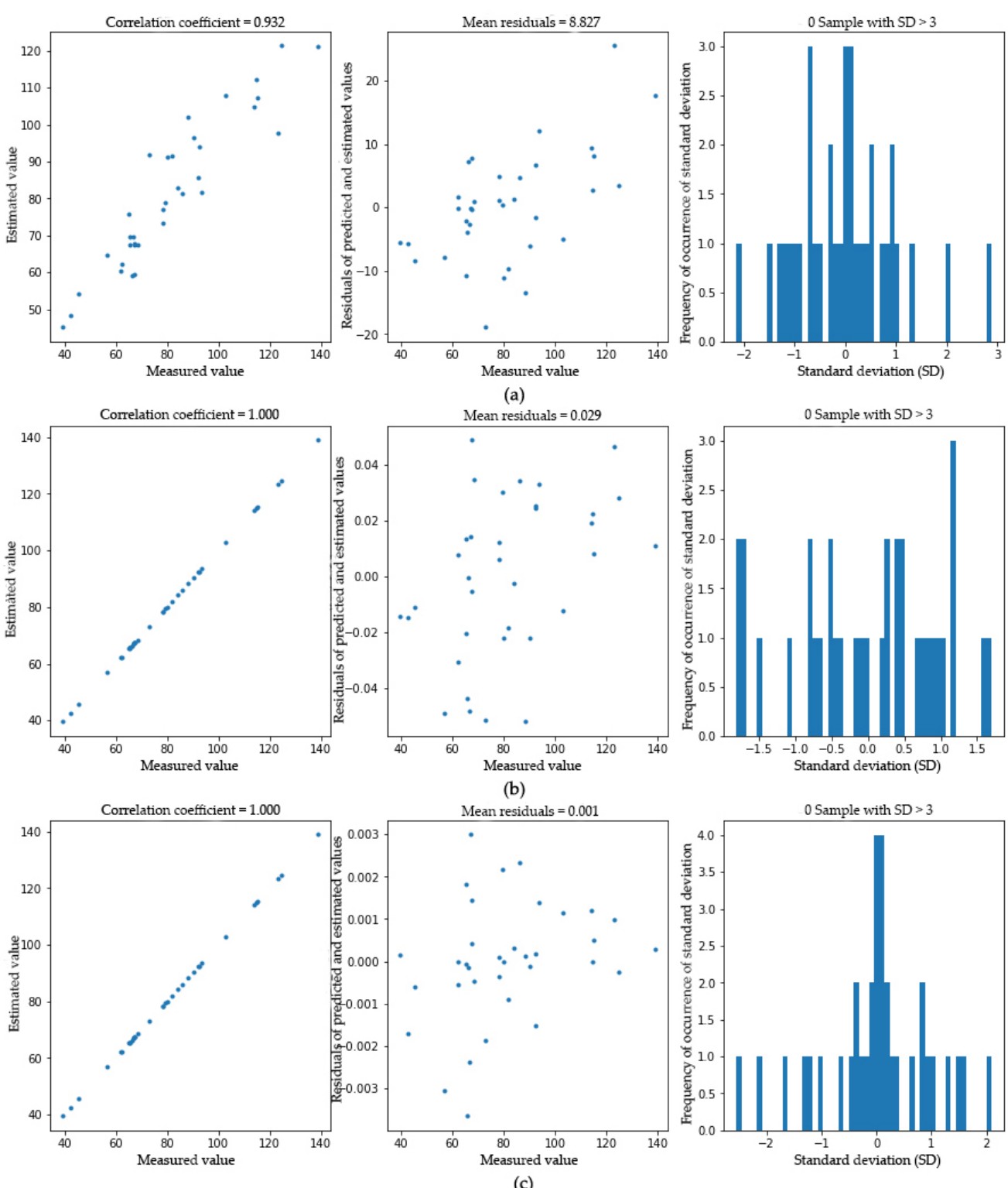

**Figure 11.** Residual and standard deviation distributions of RF, GBRT, and XGBoost model training in AM species in UAV data. (**a**) Residual and standard deviation distributions of RF model; (**b**) Residual and standard deviation distributions of GBRT model; (**c**) Residual and standard deviation distributions of XGBoost model.

Biological invasion, pests, and diseases are the main natural factors of mangrove degradation in the Guangxi area [86]. Invasion of *Sporobolus alterniflorus* decreases the carbon stock content of the soil, and *Sporobolus alterniflorus* tends to tolerate increased

salinity and flood stress better than native mangroves [87]. Mangrove plants in the Beibu Gulf region of Guangxi are short and have weak population dominance, which easily allows integrals to squeeze survival space. The Sipunculus nudus worms, Phascolosoma esculenta worms, Periophthalmus cantonensis, and Boleophthalmus chinensis proliferate in the mangrove forests, and compared with the terrestrial forests, the species diversity of the mangrove forests in Guangxi is relatively homogeneous, and the pests have almost no natural enemies, which has a great impact on the survival of the mangrove forests in Guangxi [86]. Anthropogenic factors are also important factors leading to mangrove degradation. Mangroves in coastal areas of Guangxi are mainly deforested as agricultural land, aquaculture ponds, and construction land [86]. Other risks of human activities, for example, frequent harvesting activities during the fruit ripening season, have destroyed many seedlings and young trees. Further, the degradation of mangrove forests is caused by the flushing of effluent discharged from human aquaculture ponds forming ditches that run through the area, dividing the mangroves into many isolated patches. Establishing protected areas and restoring degraded ecosystems are effective measures to conserve mangroves, and ecosystem restoration can be achieved by artificially planting species-rich mangrove saplings to increase biodiversity [88,89].

## 5. Conclusions

Mangrove ecosystems are among the most productive ecosystems in the world, and chlorophyll content is an important indicator of vegetated ecosystems. The main objective of this study is to construct an optimal CCC model at the mangrove species-scale and to investigate the sensitivity of VIs in UAV and GF-6 images. According to the results, the RF model has the best estimation accuracy, followed by the GBRT regression model, and the sensitivity of VIs calculated from UAV data is better than that of GF-6 images, and the accuracy is improved when simultaneously inverting the CCC of AC. The study showed that the sensitivity of the combined vegetation indices RI35, MDATT413, and RI35 calculated from the near-infrared band and red-edge band in the UAV data was higher for the mangrove CCC estimation, and the GF-6 data had lower sensitivity for the composed VIs than the UAV images due to the missing red-edge band. Combining the red-edge and NIR indices of UAV data and MLR, GF-6, and MLR, respectively, can achieve effective estimates of mangrove species-scale canopy chlorophyll content.

The importance of mangroves and the threats to them have long been recognized, so local and national governments and international agreements have taken action to protect them. Through this study, we found that the mangrove canopy chlorophyll content was significantly low in some of the study areas, while identifying areas of vulnerable mangrove growth to support targeted mangrove protection, restoration, and conservation decision-making. We recommend more effective protection of existing mangrove resources through the establishment of mangrove reserves and the revision of protection agreements to limit human activities in mangrove-growing areas. For areas with severe mangrove degradation, timely reforestation activities should be carried out.

The CCC values can effectively reflect the health of mangroves and provide a risk warning for mangrove ecosystems to facilitate response and decision-making for mangrove conservation and restoration. The development of the mangrove protection policy helps to restrain human activities that destroy mangrove forests from taking coercive measures, and also plays an important role in the sustainable development of mangrove resources by establishing protected areas and planting trees to restore mangrove forests through the policy.

**Author Contributions:** L.D. was responsible for the concept and methodology of the study, as well as writing the manuscript. B.C. supported the development of the methodology and the design of the experiment and contributed to the review and editing of the manuscript. L.Z. and M.Y. supported the concept development and the design of the experiment. B.F., Z.Y. and B.Z. supported the statistical and thematic graphs of the experimental results. All authors have read and agreed to the published version of the manuscript.

**Funding:** This research was funded by the Innovation Drive Development Special Project of Guangxi "China-ASEAN Earth Big Data Platform and Application Demonstration" (guikeAA20302022) and by the Natural Science Foundation of China (42001361).

**Institutional Review Board Statement:** Not applicable.

**Informed Consent Statement:** Not applicable.

**Data Availability Statement:** The data presented in this study are available on request from the corresponding author (L.Z.).

**Acknowledgments:** The authors appreciate the valuable comments and constructive suggestions from the anonymous referees and the editors who helped to improve the manuscript.

**Conflicts of Interest:** The authors declare no conflict of interest.

**Safety Affirmation for Drone Communications:** The study area of this experiment is not a no-fly zone for drones, and takeoff is permitted, and the flight strictly adhered to the limited altitude. The data collected by the drones are only used for scientific research experiments and not for other purposes [90,91].

## Appendix A

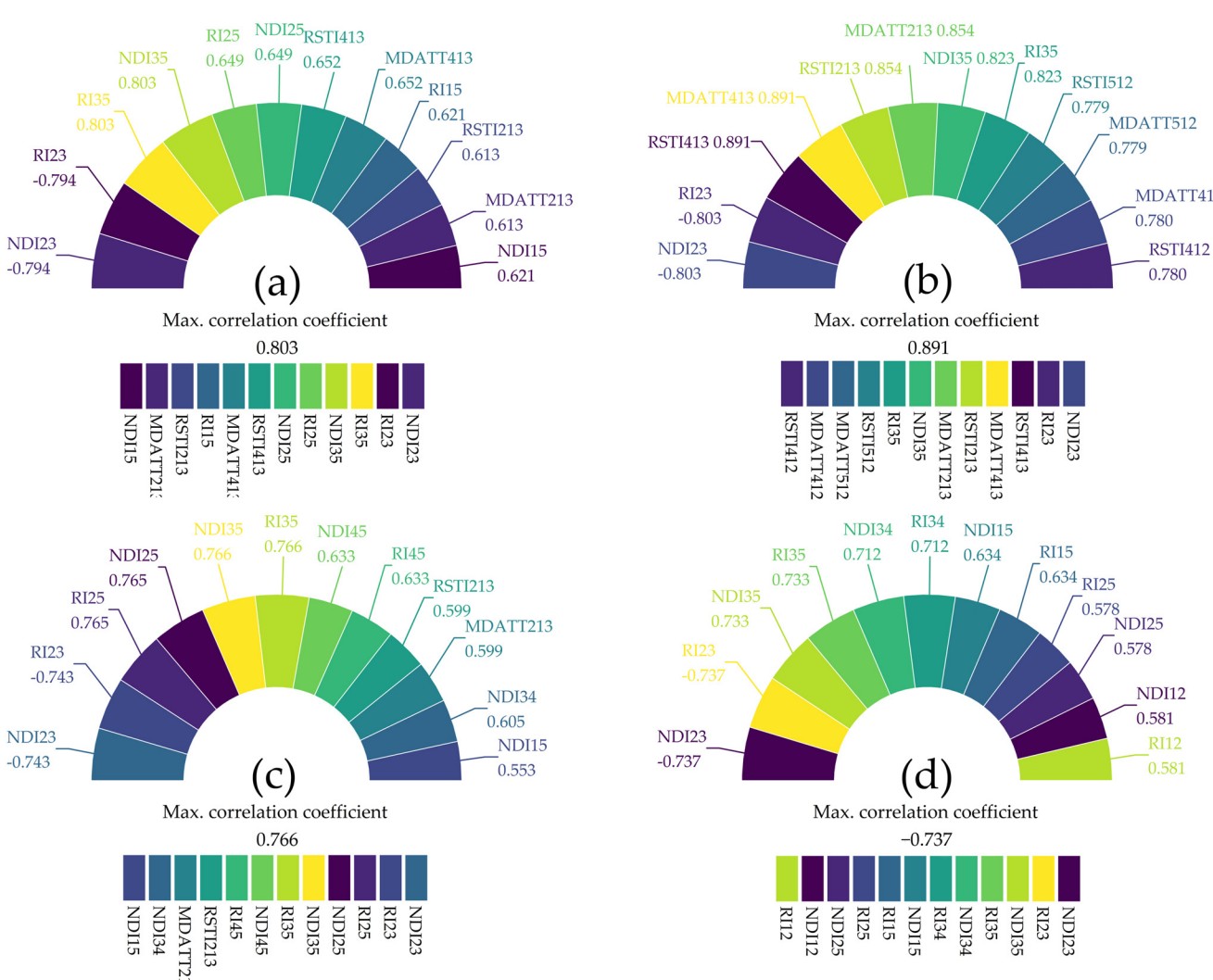

**Figure A1.** Semicircular pie charts of correlation among AM, AC, KC, and AM + AC + KC characteristic indices in UAV data. (**a**) Correlation coefficient of AM; (**b**) Correlation coefficient of AC; (**c**) Correlation coefficient of KC; (**d**) Correlation coefficient of AM + AC + KC.

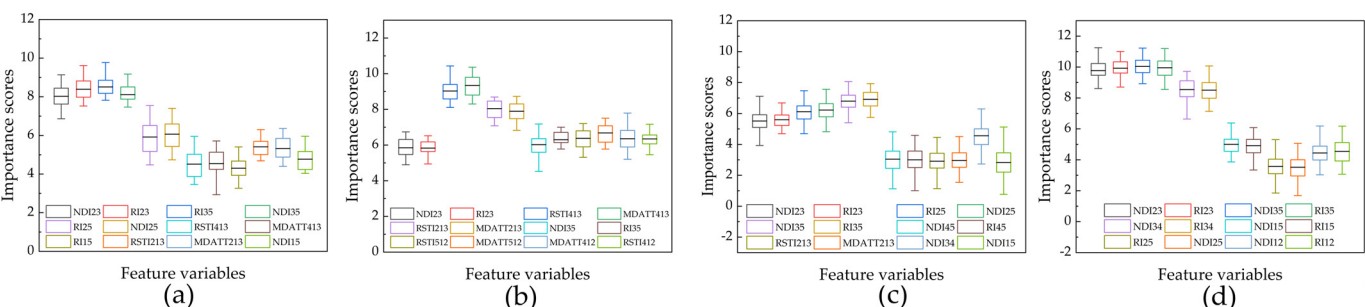

**Figure A2.** Semicircular pie charts of correlation among AM, AC, KC, and AM + AC + KC characteristic indices in GF-6 data. (**a**) Correlation coefficient of AM; (**b**) Correlation coefficient of AC; (**c**) Correlation coefficient of KC; (**d**) Correlation coefficient of AM + AC + KC.

## Appendix B

**Figure A3.** *Cont.*

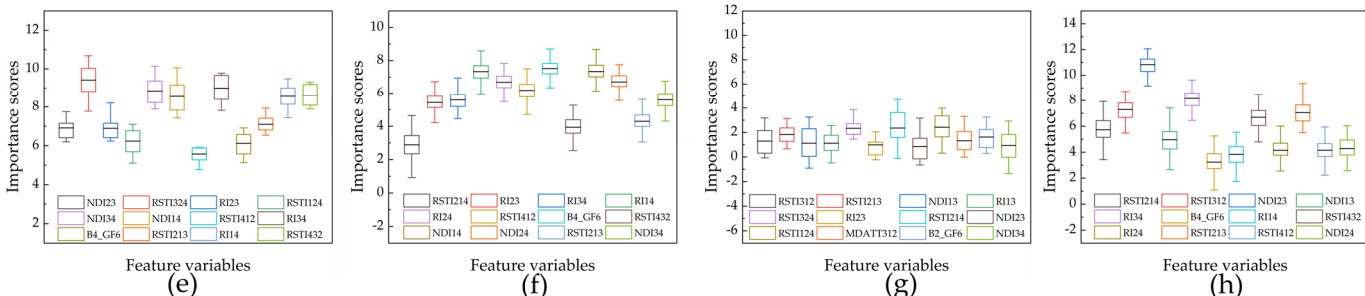

**Figure A3.** Box plot of the importance ranking of feature indices in UAV, and GF-6 data. (**a**) Importance ranking of UAV data feature variables in AM; (**b**) Importance ranking of UAV data feature variables in AC; (**c**) Importance ranking of UAV data feature variables in KC; (**d**) Importance ranking of UAV data feature variables in AM + AC + KC; (**e**) Importance ranking of GF-6 data feature variables in AM; (**f**) Importance ranking of GF-6 data feature variables in AC; (**g**) Importance ranking of GF-6 data feature variables in KC; (**h**) Importance ranking of GF-6 data feature variables in AM + AC + KC.

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
