# Peer review of "Estimation of Species-Scale Canopy Chlorophyll Content in Mangroves from UAV and GF-6 Data"

_forests, doi:10.3390/f14071417_

Round 1

Reviewer 1 Report (Previous Reviewer 1)

When compared to previous versions, the writers made significant work and improvements to the manuscript. However, I have some further suggestions for improving the paper:  

Comments and suggestions:

1. The abstract provides a good overview of the study's objectives, methods, and results. However, it could benefit from providing more context on the significance of the study and its potential implications for mangrove conservation and restoration. 

  2. Additionally, some of the sentences could benefit from more specific information on the methods used and the significance of the study's findings. Overall, the abstract could be improved by providing a more concise and informative overview of the study's main contributions to the field.

3. Could you please provide, in the form of a bulleted list, a synopsis of the most important contributions that your work has made?   4.  I recommend adding a paragraph on formal approaches for AI-based technique verification to this study to improve its quality and impact. Formal approaches, which utilize mathematical models and logic to check system correctness, are increasingly significant in AI-based technique development and validation.
5. Some relevant references related to this topic that the authors may want to consider include:

a. https://ieeexplore.ieee.org/abstract/document/9842406

b. https://incose.onlinelibrary.wiley.com/doi/abs/10.1002/inst.12434   6. In Section 4, additional clarification is needed regarding the particular connections between the dangers and factors discussed in the preceding paragraph and the findings of the study concerning the mangrove health indicator parameters.

7. In addition, the section might be strengthened by the addition of further information regarding the particular policy proposals and tactics that can be implemented in order to create protected areas and restore ecosystems that have been damaged.

  8. The conclusion section provides a clear and concise summary of the study's main findings and their implications for mangrove conservation and restoration. However, it could benefit from more explanation on how the study's findings relate to the larger context of mangrove conservation and the challenges faced in protecting and restoring mangrove ecosystems.

9. Additionally, the section could benefit from more detail on the specific policy recommendations and strategies that can be used to implement the study's findings in practice.

10. Overall, the conclusion section could be improved by providing a stronger connection between the study's main findings and their practical implications for mangrove conservation and restoration.

Can be improved.

Author Response

Manuscript: forests-2428753

Title: Estimation of Specie-scale Canopy Chlorophyll Content in Mangroves from UAV and GF-6 Data

We appreciate the valuable comments and suggestions from the editors and reviewers. We have revised the paper carefully and made corrections accordingly. Revised portions of the manuscript are highlighted in red color. Our detailed point-by-point responses to the reviewers' comments are given below. 

Point 1: The abstract provides a good overview of the study's objectives, methods, and results. However, it could benefit from providing more context on the significance of the study and its potential implications for mangrove conservation and restoration.

Response 1: We added background on the significance of the study and its potential impact on mangrove conservation and restoration. (See Lines: 12-15)

Point 2: Additionally, some of the sentences could benefit from more specific information on the methods used and the significance of the study's findings. Overall, the abstract could be improved by providing a more concise and informative overview of the study's main contributions to the field.    

Response 2: We revised the abstract and provided a concise and informative overview of the main contributions in this field. (See Lines: 38-39)

Point 3: Could you please provide, in the form of a bulleted list, a synopsis of the most important contributions that your work has made?

Response 3: The most important contributions of this study are as follows:

  • Besides the existing studies, we have tested GF-6 data, which is quite new for estimating species-scale canopy chlorophyll content in mangroves.
  • We have thoroughly investigated how different machine learning algorithms behaves under different circumstances towards the establishment of canopy chlorophyll content.
  • We used a combination of UAV and GF-6 from the view of spaceborne and airborene platform for decsion makers to balacnce the cost and efficiency.

Point 4: I recommend adding a paragraph on formal approaches for AI-based technique verification to this study to improve its quality and impact. Formal approaches, which utilize mathematical models and logic to check system correctness, are increasingly significant in AI-based technique development and validation.

Response 4: Our present study is mainly focused on machine learning frameworks and we will focus more on the AI-based technique in future works.

Point 5: Some relevant references related to this topic that the authors may want to consider include:

  1. https://ieeexplore.ieee.org/abstract/document/9842406
  2. https://incose.onlinelibrary.wiley.com/doi/abs/10.1002/inst.12434

Response 5: We included in the discussion section an outlook on formal approaches for AI-based technique verification and cited these references (See Lines: 609-612)

Point 6: In Section 4, additional clarification is needed regarding the particular connections between the dangers and factors discussed in the preceding paragraph and the findings of the study concerning the mangrove health indicator parameters.

Response 6: We have added a short paragraph discussing the specific link between risk factors and the results of the mangrove canopy chlorophyll content health indicator parameter study. (See Lines: 624-627)

Point 7: In addition, the section might be strengthened by the addition of further information regarding the particular policy proposals and tactics that can be implemented in order to create protected areas and restore ecosystems that have been damaged.

Response 7: We have added relevant information regarding the local policy and measurements. (See Lines: 627-636)

Point 8: The conclusion section provides a clear and concise summary of the study's main findings and their implications for mangrove conservation and restoration. However, it could benefit from more explanation on how the study's findings relate to the larger context of mangrove conservation and the challenges faced in protecting and restoring mangrove ecosystems.

Response 8: Revised as suggestions. Thanks for your helpful comments, it would be interesting to apply this to a large area and we have revised the content to further discuss this iussee. (See Lines: 672-677)

Point 9: Additionally, the section could benefit from more detail on the specific policy recommendations and strategies that can be used to implement the study's findings in practice.

Response 9: We have provided the scope from the view of policy (See Lines: 677-680)

Point 10: Overall, the conclusion section could be improved by providing a stronger connection between the study's main findings and their practical implications for mangrove conservation and restoration.

Response 10: Revised as suggestions. 

Thanks for your helpful suggestions and we have improved the conclusion part. (See Lines: 672-680)

Reviewer 2 Report (New Reviewer)

General comments:

The manuscript evaluated the capability of UAV, GF-6 data, and machine learning regression algorithms in estimating mangrove species-scale CCC. Overall, The manuscript is clear and well-written, as well graphical representations are of high quality.

However, there are considerable limitations that would have to be addressed. See comments below for details.

Main concerns:

1.      The second paragraph of the introduction is a little lengthy. I suggest there is no need to discuss too much in Sentinel-2 data. You only need to express why you use UAV and GF-6 data.

2.      In the introduction part, I suggest adding a small paragraph (behind the third paragraph) : the research progress and existing problems of mangrove CCC using UAV or satellite data. This can better reflect your research to fill these research gaps, highlighting the necessity and importance of your research.

3.      SPAD value is no physical unit, you have converted it to ug/cm2? Why does LAI * LCC become a physical unit (ug/cm2)?

Specific comments:

L14: unmanned aerial vehicle (UAV) imagery, Gaofen-6 (GF-6) satellite imagery

L16: “Sporobolus alterniflorus” should be in italic?

L26-27: This word is in italic. “Aegiceras corniculatum

L134: Add a related reference: Guo Y, Chen S, Li X, et al. Machine learning-based approaches for predicting SPAD values of maize using multi-spectral images[J]. Remote Sensing, 2022, 14(6): 1337.

L175: (a) Study area and (b)….

L245: The font type of “flowchart” needs to be modified.

L503-504: Add a reference: Wu L, Wang L, Shi C, et al. Detecting mangrove photosynthesis with solar-induced chlorophyll fluorescence[J]. International Journal of Remote Sensing, 2022, 43(3): 1037-1053.

Author Response

Manuscript: forests-2428753

Title: Estimation of Specie-scale Canopy Chlorophyll Content in Mangroves from UAV and GF-6 Data

We appreciate the valuable comments and suggestions from the editors and reviewers. We have revised the paper carefully and made corrections accordingly. Revised portions of the manuscript are highlighted in red color. Our detailed point-by-point responses to the reviewers' comments are given below.

Main concerns:

Point 1: The second paragraph of the introduction is a little lengthy. I suggest there is no need to discuss too much in Sentinel-2 data. You only need to express why you use UAV and GF-6 data.

Response 1: Thanks for your suggestion and we have shorten the part for Sentinel-2 data as suggested.

Point 2: In the introduction part, I suggest adding a small paragraph (behind the third paragraph): the research progress and existing problems of mangrove CCC using UAV or satellite data. This can better reflect your research to fill these research gaps, highlighting the necessity and importance of your research.

Response 2: We added a short paragraph (after the third paragraph of the introduction) to describe the progress and problems of research, the necessity, and importance of research on mangrove CCC estimation from UAV or satellite data. (See Lines:146-155)

Point 3: SPAD value is no physical unit, you have converted it to ug/cm2? Why does LAI * LCC become a physical unit (ug/cm2)?

Response 3: We have replaced the μg/cm2 units in the text with SPAD. (where Figure 7- Figure 10 is also modified)

Specific comments:

Point 4: L14: unmanned aerial vehicle (UAV) imagery, Gaofen-6 (GF-6) satellite imagery

Response 4: Revised.

Point 5: L16: “Sporobolus alterniflorus” should be in italic?

Response 5: The Latin name of the species is written uniformly in italics.

Point 6: L26-27: This word is in italic. “Aegiceras corniculatum”

Response 6: The Latin name of the species is written uniformly in italics.

Point 7: L134: Add a related reference: Guo Y, Chen S, Li X, et al. Machine learning-based approaches for predicting SPAD values of maize using multi-spectral images[J]. Remote Sensing, 2022, 14(6): 1337.

Response 7: Added as suggestions.

Point 8: L175: (a) Study area and (b)….

Response 8: Revised. (See Line: 186)

Point 9: L245: The font type of “flowchart” needs to be modified.

Response 9: Revised.

Point 10: L503-504: Add a reference: Wu L, Wang L, Shi C, et al. Detecting mangrove photosynthesis with solar-induced chlorophyll fluorescence[J]. International Journal of Remote Sensing, 2022, 43(3): 1037-1053.

Response 10: Added as suggestions. (See Lines: 514-519)

Reviewer 3 Report (New Reviewer)

General comments

The manuscript deals with the evaluation of the capability of UAV, GF-6 data, and machine learning regression algorithms to estimate mangrove species-scale canopy chlorophyll content (CCC). The topic is interesting, the manuscript well-structured, and the findings helpful. However, some points need improvement.

Specific comments
Line 188. Please, add the initial letters of the plant names in order to correspond with the initials written in Table 1.

Table 1. Please, replace “Measured average” with Mean value. Also, add the standard error of mean after each value.

Line 240. Please, delete “In this study,”

Lines 302-303, Please, delete the text “The coefficient of determination (R²) is used to indicate the degree of fit of the regression curve to the measured values, and the value of R² is in [0, 1], and the larger R² 3indicates the better fit. In this paper,”

Line 363. Please, start with capital letter after dot.

Line 479. Please, delete “In this paper,”

Lines 604-609, and elsewhere. Please, write the latin names of the species in italics.

Subsection, 4.3. Mangrove risk prevention and protection measures
Please, enrich the subsection with more details on Mangrove risk prevention and protection measures. Only the last sentence “Establishing protected areas and ……………………..to increase biodiversity [87,88]”, deals with measures.

Conclusion
Please, Details such as “…and the R² accuracy is improved by 0.152 and the RMSE is reduced by 4.735 630 μg/cm²”, should be removed from the Conclusion section.

General comments

The manuscript deals with the evaluation of the capability of UAV, GF-6 data, and machine learning regression algorithms to estimate mangrove species-scale canopy chlorophyll content (CCC). The topic is interesting, the manuscript well-structured, and the findings helpful. However, some points need improvement.

Specific comments
Line 188. Please, add the initial letters of the plant names in order to correspond with the initials written in Table 1.

Table 1. Please, replace “Measured average” with Mean value. Also, add the standard error of mean after each value.

Line 240. Please, delete “In this study,”

Lines 302-303, Please, delete the text “The coefficient of determination (R²) is used to indicate the degree of fit of the regression curve to the measured values, and the value of R² is in [0, 1], and the larger R² 3indicates the better fit. In this paper,”

Line 363. Please, start with capital letter after dot.

Line 479. Please, delete “In this paper,”

Lines 604-609, and elsewhere. Please, write the latin names of the species in italics.

Subsection, 4.3. Mangrove risk prevention and protection measures
Please, enrich the subsection with more details on Mangrove risk prevention and protection measures. Only the last sentence “Establishing protected areas and ……………………..to increase biodiversity [87,88]”, deals with measures.

Conclusion
Please, Details such as “…and the R² accuracy is improved by 0.152 and the RMSE is reduced by 4.735 630 μg/cm²”, should be removed from the Conclusion section.

Author Response

Manuscript: forests-2428753

Title: Estimation of Specie-scale Canopy Chlorophyll Content in Mangroves from UAV and GF-6 Data

We appreciate the valuable comments and suggestions from the editors and reviewers. We have revised the paper carefully and made corrections accordingly. Revised portions of the manuscript are highlighted in red color. Our detailed point-by-point responses to the reviewers' comments are given below.

Point 1: Line 188. Please, add the initial letters of the plant names in order to correspond with the initials written in Table 1.

Response 1: Revised.

Point 2: Table 1. Please, replace “Measured average” with Mean value. Also, add the standard error of mean after each value.

Response 2:  Revised.

Point 3: Line 240. Please, delete “In this study,”

Response 3: Revised.

Point 4: Lines 302-303, Please, delete the text “The coefficient of determination (R²) is used to indicate the degree of fit of the regression curve to the measured values, and the value of R² is in [0, 1], and the larger R² 3indicates the better fit. In this paper,”

Response 4: Revised as suggestions.

Point 5: Line 363. Please, start with capital letter after dot.

Response 5: Revised.

Point 6: Line 479. Please, delete “In this paper,”

Response 6: Revised.

Point 7: Lines 604-609, and elsewhere. Please, write the latin names of the species in italics.

Response 7: Revised.

Point 8: Subsection, 4.3. Mangrove risk prevention and protection measures. Please, enrich the subsection with more details on Mangrove risk prevention and protection measures. Only the last sentence “Establishing protected areas and ……………………..to increase biodiversity [87,88]”, deals with measures.

Response 8: Revised as suggestions. (See Lines: 629-636)

Point 9: Conclusion:Please, Details such as “…and the R² accuracy is improved by 0.152 and the RMSE is reduced by 4.735 630 μg/cm²”, should be removed from the Conclusion section.

Response 9: Revised. (See Lines: 664)

This manuscript is a resubmission of an earlier submission. The following is a list of the peer review reports and author responses from that submission.

Round 1

Reviewer 1 Report

Summary/Contribution: "In this study, the authors investigated the capability of UAV data and GF-6 data for estimating specie-scale canopy chlorophyll content (CCC) in mangroves."   Suggestions/Comments: 1. The title is too long and needs to be shortened. In addition, the authors are invited to use the capitalized form for the title.   2. Use commas to separate the numbers that show the affiliations of the authors.   3. The abstract contains many abbreviations and many numbers. It is preferable to avoid this.   4. The introduction needs to be split into two sections: Introduction + Related works.   5. The contributions of the authors need to be clearly summarized in the introduction.   6. Line 208: "In this paper, GF-6 data were acquired through the China Resources" ===> the authors are invited to provide more arguments about the choice of this dataset.   7. Line 227-229: Similarly, the authors are invited to argue more about the adoption of "Random forest (RF), gradient boosting (GBRT), and extreme gradient 228 boosting (XGBoost) algorithms"   8. More explanation about equations 1 and 2 is needed.   9. The font used in Figures 4 and 5 is too small and needs to be enlarged.   10. Some of the figures may be moved to the appendix.   11. The authors need to add a short paragraph about the security aspects of drone communications.    12. For this purpose, they may include the following references (and others) in their study:    a. https://ieeexplore.ieee.org/document/9842403    b. https://www.mdpi.com/1424-8220/21/6/2057

Author Response

Response to Reviewer 1 Comments

Manuscript: forests-2292461

Title: Estimation of Specie-scale Canopy Chlorophyll Content in Mangroves from UAV and GF-6 Data

We appreciate the valuable comments and suggestions from the editors and reviewers. We have revised the paper carefully and made corrections accorddingly. Revised portions of the manuscript are highlighted in red color. Our detailed point-by-point responses to the reviewers' comments are given below.

Point 1: The title is too long and needs to be shortened. In addition, the authors are invited to use the capitalized form for the title.  

Response 1: We have shortened and revised the paper's title as ‘Estimation of Specie-scale Canopy Chlorophyll Content in Mangroves from UAV and GF-6 Data’.

Point 2: Use commas to separate the numbers that show the affiliations of the authors. 

Response 2: Revised.

Point 3: The abstract contains many abbreviations and many numbers. It is preferable to avoid this.

Response 3: Revised as suggestions to make it more concise.

Point 4: The introduction needs to be split into two sections: Introduction + Related works.   

Response 4: We have divided the introduction as suggested. The first paragraph is the introduction, the second paragraph is the related works on remote sensing data for vegetation parameter estimation, the third paragraph is the related works on empirical inversion algorithms for vegetation parameters, and the fourth paragraph stated the objective of the manuscript. (See Lines: 53-65)

Point 5: The contributions of the authors need to be summarized in the introduction.

Response 5: The authors' contributions are addressed in the last paragraph of the introduction. (See Lines: 146-158)

Point 6: Line 208: "In this paper, GF-6 data were acquired through the China Resources" ===> the authors are invited to provide more arguments about the choice of this dataset.

Response 6:

We chose this data based on:

1) GF-6 data is relatively new to China and there are fewer related studies;

2) the previous inversion parameters of the high-fraction series of satellites have achieved good results, and this test is a new satellite;

3) There are also relatively few studies on canopy chlorophyll content, and GF-6 data can make some useful attempts.

Meanwhile, We added a corresponding study of GF-6 data in vegetation parameter extraction in the introduction section, demonstrating that GF-6 satellite data could invert vegetation parameters. The application of GF-6 satellite data to the inversion of canopy chlorophyll content at the species scale of mangrove forests in Beibu Bay, Guangxi is the first attempt of the authors. The following are references to relevant studies applying GF-6 satellite data to inverse vegetation parameters: (See Lines: 215-218)

References:

Sun Y, Wang B, Zhang Z. Improving leaf area index estimation with chlorophyll insensitive multispectral red-edge vegetation indices. IEEE. J. Sel. Top. Appl. Earth Obs. Remote. Sens. 2023.

Jiang X, Fang S, Huang X, et al. Rice mapping and growth monitoring based on time series GF-6 images and red-edge bands. Remote. Sens. 2021, 13(4): 579.

Chen Z, Jia K, Wei X, et al. Improving leaf area index estimation accuracy of wheat by involving leaf chlorophyll content information. COMPUT. ELECTRON. AGR. 2022, 196: 106902.

Chen L, Letu H, Fan M, et al. An Introduction to the Chinese High-Resolution Earth Observation System: Gaofen-1~ 7 Civilian Satellites. Int. J. Remote Sens. 2022, 2022.

Point 7: Line 227-229: Similarly, the authors are invited to argue more about the adoption of "Random forest (RF), gradient boosting (GBRT), and extreme gradient 228 boosting (XGBoost) algorithms"

Response 7: We have added discussionsfor the effective inversion of vegetation parameters by RF, GBRT, and XGBoost algorithms, with corresponding references for justification. More references are as follows: (See Lines: 235-236)

References:

Fu B, Sun J, Wang Y, et al. LAI estimation of mangrove communities using DLR algorithm and sample enhancement with UAV, hyperspectral and SAR images. Front. Mar. Sci. 2022, 1178.

Zhen J, Jiang X, Xu Y, et al. Mapping leaf chlorophyll content of mangrove forests with Sentinel-2 images of four periods. Int. J. Appl. Earth Obs. Geoinf. 2021, 102: 102387.

Pham T D, Yokoya N, Xia J, et al. Comparison of machine learning methods for estimating mangrove above-ground biomass using multiple source remote sensing data in the red river delta biosphere reserve, Vietnam. Remote. Sens. 2020, 12(8): 1334.

Miao J, Zhen J, Wang J, et al. Mapping Seasonal Leaf Nutrients of Mangrove with Sentinel-2 Images and XGBoost Method. Remote. Sens. 2022, 14(15): 3679.

Point 8: More explanation about equations 1 and 2 is needed. 

Response 8: We have added more explanations for equation 1 and equation 2. (See Lines: 309-316, and Lines: 309-316)

Point 9: The font used in Figures 4 and 5 is too small and needs to be enlarged.

Response 9: Revised.

Point 10: Some of the figures may be moved to the appendix.

Response 10: We have moved Figures 5 and 6 in the original manuscript to Appendix A and B.

Point 11: The authors need to add a short paragraph about the security aspects of drone communications.

Response 11: We have added a short paragraph about the security aspects of drone communications. (See Lines: 662-665)

Point 12: For this purpose, they may include the following references (and others) in their study:    a. https://ieeexplore.ieee.org/document/9842403    b. https://www.mdpi.com/1424-8220/21/6/2057.

Response 12: In the text, we have quoted them. (See Lines: 662-665)

Reviewer 2 Report

This paper aims to design a prediction model of specie-scale canopy chlorophyll content (CCC) in mangroves by using UAV data and GF-6 data.  It is an interesting topic for CCC prediction. Moreover, for the applicability of RF, GBRT, and XGBoost algorithms in mangrove CCC estimation is still limited, these algorithms were used as the classifiers for the prediction model. It would be useful to improve the accuracy of the model.  However, experimental results showed that RF achieved optimal results in inverse mangrove AC species CCC with higher stability and robustness.  The authors should explained the reason that GBRT, and XGBoost did not obtained better accuracy than RF.

Author Response

Response to Reviewer 2 Comments

Manuscript: forests-2292461

Title: Estimation of Specie-scale Canopy Chlorophyll Content in Mangroves from UAV and GF-6 Data

We appreciate the valuable comments and suggestions from the editors and reviewers. We have revised the paper carefully and made corrections accorddingly. Revised portions of the manuscript are highlighted in red color. Our detailed point-by-point responses to the reviewers' comments are given below.

Point 1: The authors should explained the reason that GBRT, and XGBoost did not obtained better accuracy than RF.

Response 1: We include a paragraph to explain why RF's accuracy is better than GBRT and XGBoost. We added experiments on model training to visualize the predictive performance of the training data during the training of RF, GBRT, and XGBoost models. (See Lines: 576-589)

Reviewer 3 Report

The proposed research presents the application of three machine-learning regressor models for estimating canopy chlorophyll content in mangroves. The authors compared the performance of regressor models when using UAV and satellite imagery. The outcomes show that UAV imagery is more suitable than satellite imagery for evaluating the canopy chlorophyll content of mangroves. Specifically, the best regression performance is obtained by random forest regressor and UAV images (R^2=0.8, rmse=15.45).

The paper is generally well structured, yet the sentences should be rewritten. Several instances exist where the sentences need to be shorter, difficulting the reading and understanding process. Further, one of my primary concerns is about the discussion of results. Specifically, this section should describe the results as well error's source. Nevertheless, its composition makes it feel more like an introduction than a discussion section, for instance, from lines 562 - 570.
Furthermore, sentences might not be supported by the proposed methodology or results. For example, from lines 548 to 552, the authors discussed that XGboost is prone to overfitting although the enhancement of the XGboost algorithm. However, I could not find the methodology subsection where this improvement occurs. I am aware that authors have used that lines to demonstrate that XGboost is susceptible to overfitting, which might not be suitable. Nonetheless, the results section needs to clearly show the overfitting behavior.

Particular concerns are as follows:

### Introduction
What are the unsolved issues of state-of-the-art works that your work expects to address?

It is essential to highlight that several works demonstrate that UAV imagery is better placed for vegetation characterization than satellite imagery. What is the contribution of your work when studying the UAV and GF-6 multispectral data for mangrove chlorophyll estimation?

The general audience might not be familiar with the term "invert" or "inversion" please add a description of what you refer to with that term.

Lines 24-27: The sentence is too long. I suggest rewriting it. In general, there exist several instances of long sentences; please rewrite them.

Lines 28-31: These lines might be better placed at the beginning of the abstract; they sound like motivation. I suggest rewriting these lines, considering that the abstract's general structure is: motivation, problem statement, proposed solution, and outcomes of the proposed strategy.

Lines 60-73: This paragraph disrupts the reading flow. The CCC term is introduced in the last paragraph, then these lines talk about mangroves, and once again, the CCC term is presented. I suggest moving this paragraph (it could be used as motivation for the current work).

Lines 83-88: Hard to follow; sentence needs to be shorter.

Lines 109-110: Add references that support that statement.

Line 148: Check these words: differnet, pecies, quanlity

### Material and methods

I suggest describing which variables are used as input and target variables in the methods sections. From the reading, I could infer that the target variable is the CCC, yet it should be explicit in the text for easy reading flow.

Moreover is quite ambiguous if the pixel values of each image are used as input to the machine learning regressors. Please add a clear description of it.

Figure 1: Sort subfigures; the subfigure d should be plated at the right-bottom side (suggester order: left to right and from top to bottom)  

Line 176: square meters?

Line 241: what are the characteristic variables? Do you refer to the reflectance in each spectral band?

Line 279: Do not start with R^2. I suggest using: "The R^2."

### Results
In general, redundant figures show the same outcomes in two different formats. Leave just the most informative figure to avoid the population of figures. Furthermore, the text in the images is hard to read; please increase the font of the images' text.

Line 296: What are feature parameters? Do they are the vegetation indices from Table 2?

Figure 4: What do "23", "25", and "213" numbers mean? They need to be explained in the text.

### Discussions

Line 546: What is the multisource data? Which results show the multisource data outcomes?

Lines 562-570: These lines might be better placed in the introduction section.

### Conclusions
Lines 605-607: Where are the results showing the combination of UAV and GF-6 data?

Author Response

Response to Reviewer 3 Comments

Manuscript: forests-2292461

Title: Estimation of Specie-scale Canopy Chlorophyll Content in Mangroves from UAV and GF-6 Data

We appreciate the valuable comments and suggestions from the editors and reviewers. We have revised the paper carefully and made corrections accorddingly. Revised portions of the manuscript are highlighted in red color. Our detailed point-by-point responses to the reviewers' comments are given below.

Point 1: The paper is generally well structured, yet the sentences should be rewritten. Several instances exist where the sentences need to be shorter, difficulting the reading and understanding process. Further, one of my primary concerns is about the discussion of results. Specifically, this section should describe the results as well error's source. Nevertheless, its composition makes it feel more like an introduction than a discussion section, for instance, from lines 562 - 570.

Response 1: The author has rewritten this section. (See Lines: 606-623).

Point 2: Furthermore, sentences might not be supported by the proposed methodology or results. For example, from lines 548 to 552, the authors discussed that XGboost is prone to overfitting although the enhancement of the XGboost algorithm. However, I could not find the methodology subsection where this improvement occurs. I am aware that authors have used that lines to demonstrate that XGboost is susceptible to overfitting, which might not be suitable. Nonetheless, the results section needs to clearly show the overfitting behavior.

Response 2:  The authors' improvements to the RF, GBRT, and XGBoost models are mainly model tuning by setting the corresponding parameters. In the revised manuscript section, the authors added a subsection on model parameter tuning. (See Lines: 283-292)

For different mangrove species, some machine learning regression models may be overfitted, for example, when estimating the CCC of AM species from UAV data, RF achieved better estimation results, while the accuracy of XGBoost and GBRT was lower than that of RF. To visualize the model fitting situation, the authors supplemented the model training experiments, and from the experimental results, it is clear that it is the GBRT and XGBoost models that show overfitting and lead to lower prediction accuracy. (See Lines: 606-623)

### Introduction

Point 3: What are the unsolved issues of state-of-the-art works that your work expects to address?

Response 3: The unresolved issues of state-of-the-art works are: 1) For a long time, less attention has been paid to the health status of mangrove forests and a lack of relevant studies. The inversion of canopy chlorophyll content, which is an important indicator for assessing the health of mangroves, still needs to be studied in depth (Lines: 59-64). 2) The reliability of using UAV and GF-6 multispectral data as the data source for mangrove species-scale canopy chlorophyll content inversion still needs to be further explored (Lines: 110-113). 3) The applicability of RF, GBRT, and XGBoost algorithms in mangrove CCC estimation, especially at the species scale, is still very limited (See Lines: 140-143).

The expectations of our work are: 1)we quantitatively invert the canopy chlorophyll content in the Beibu Gulf region of Guangxi and assess the health status of the region based on the inversion results; 2)we evaluate the ability of UAV data and GF-6 data to estimate species-scale canopy chlorophyll content (CCC) in mangrove forests in the Beibu Gulf region of Guangxi; 3)we estimate the species-scale canopy chlorophyll content (CCC) based on correlation coefficients and characteristic variable selection methods to select the optimal variables, we used RF, GBRT, and XGBoost machine learning regression algorithms to achieve high inversion accuracy (See Lines: 146-152).

Point 4: It is essential to highlight that several works demonstrate that UAV imagery is better placed for vegetation characterization than satellite imagery. What is the contribution of your work when studying the UAV and GF-6 multispectral data for mangrove chlorophyll estimation?

Response 4: The advantages of UAV data over satellite imagery have been proven (See Lines: 84-95).

In this study, our contributions were: This study provides a new idea for mangrove health monitoring by estimating the physicochemical parameters of mangroves. It provides a vital data reference for mangrove restoration and conservation in the Beibu Gulf region of Guangxi, China. It provides risk warnings for mangrove ecosystems to facilitate rapid response and decision-making for mangrove conservation and restoration (See Lines: 152-157).

Point 5: The general audience might not be familiar with the term "invert" or "inversion" please add a description of what you refer to with that term.

Response 5: The terms "invert" or "inversion" are used synonymously with "estimate" and "retrieve" in this paper. The term "invert" or "inversion" is used in many papers to express the meaning of remote sensing estimation, for example, in the following papers. We have described the words in the paper. (See Lines: 109).

References:

Atzberger, C., Darvishzadeh, R., Immitzer, M.,Schlerf, M.,Skidmore, A., Maire, G. Comparative analysis of different retrieval methods for mapping grassland leaf area index using airborne imaging spectroscopy. Int. J. Appl. Earth Observ. Geoinf. 2015, 43, 19–31.

Alonzo, M., Bookhagen, B., McFadden, J.P., Sun, A., Roberts, D. A. 2015. Mapping urban forest leaf area index with airborne lidar using penetration metrics and allometry. Remote Sens.Environ. 2015, 162, 141–153.

Point 6: Lines 24-27: The sentence is too long. I suggest rewriting it. In general, there exist several instances of long sentences; please rewrite them.

Response 6: Revised (See Lines: 27-31).

Point 7: Lines 28-31: These lines might be better placed at the beginning of the abstract; they sound like motivation. I suggest rewriting these lines, considering that the abstract's general structure is: motivation, problem statement, proposed solution, and outcomes of the proposed strategy.

Response 7: Revised (See Lines: 11-14).

Point 8: Lines 60-73: This paragraph disrupts the reading flow. The CCC term is introduced in the last paragraph, then these lines talk about mangroves, and once again, the CCC term is presented. I suggest moving this paragraph (it could be used as motivation for the current work).

Response 8: Revised as suggestions to make it easier to read (See Lines: 52-61).

Point 9: Lines 83-88: Hard to follow; sentence needs to be shorter.

Response 9: Revised as suggestions to make it easier to read (See Lines: 73-79).

Point 10: Lines 109-110: Add references that support that statement.

Response 10: The authors have added the appropriate references to support this statement. The references are as follows: (See Lines: 110-113).

References:

Wang, L., Jia, M., Yin, D., Tian, J. A review of remote sensing for mangrove forests: 1956–2018. Remote Sens. Environ. 2019, 231, 111223.

It is mentioned in the literature that the detailed information of mangroves cannot be distinguished due to the limitation of image spatial resolution, and the emergence of platforms such as high-resolution satellites provides the possibility of LAI inversion. And it is mentioned in the paper that 64 published studies on mapping LAI have focused on exploring the potential of different types of remote sensing sensors, including high-resolution satellite images and UAV multispectral images, etc. GF-6 data belong to high-resolution satellite images, and UAV has an ultra-high resolution, which is worth exploring in the applicability of estimation of vegetation parameters. Regarding the canopy chlorophyll content, which is an important indicator of vegetation health as LAI, there is little literature on the application of high-resolution imagery to invert the species-scale canopy chlorophyll content of mangroves, and its related research is of great importance.

Point 11: Line 148: Check these words: differnet, pecies, quanlity

Response 11: These words were spelled incorrectly in the original manuscript and have been corrected by the author: “different”,  “species”,  “quantify” (See Lines: 148-152).

### Material and methods

Point 12: I suggest describing which variables are used as input and target variables in the methods sections. From the reading, I could infer that the target variable is the CCC, yet it should be explicit in the text for easy reading flow.

Response 12: The input variables of the machine learning regression model in this paper are reflectance bands and vegetation indices after correlation coefficient screening and feature variable selection, and the target variable is CCC (See Lines: 235-238).

Point 13: Moreover is quite ambiguous if the pixel values of each image are used as input to the machine learning regressors. Please add a clear description of it.

Response 13: The authors explain the input parameters of the machine learning regression model, i.e., the feature variables (reflectance bands or combined vegetation indices sensitive to CCC obtained after correla-tion analysis and feature variable selection), instead of directly using the pixel values of each image as the input to the machine learning regressor (See Lines: 235-238).

Point 14: Figure 1: Sort subfigures; the subfigure should be plated at the right-bottom side (suggester order: left to right and from top to bottom) 

Response 14: Revised as suggestions.

Point 15: Line 176: square meters?

Response 15: The term "collection area" in the original manuscript is ambiguous, so the author changed it to “survey site”, and the unit is still "m" (See Lines: 183-184).

Point 16: Line 241: what are the characteristic variables? Do you refer to the reflectance in each spectral band?

Response 16: "Characteristic variables" means: reflectance bands or combined vegetation indices sensitive to CCC obtained after correla-tion analysis and feature variable selection. The authors have annotated the first paragraph of the methods section. To avoid ambiguity, the authors have revised the term "characteristic variables" to " feature variables".

Point 17: Line 279: Do not start with R^2. I suggest using: "The R^2."

Response 17: Revised.

### Results

Point 18: In general, redundant figures show the same outcomes in two different formats. Leave just the most informative figure to avoid the population of figures. Furthermore, the text in the images is hard to read; please increase the font of the images' text.

Response 18: The authors have placed the figures with the same results in the appendix, and have also enlarged the font in the corresponding figures.

Point 19: Line 296: What are feature parameters? Do they are the vegetation indices from Table 2?

Response 19: The term "feature parameters" has the same meaning as "feature variables" in the text, which has been changed to "feature variables".

Point 20: Figure 4: What do "23", "25", and "213" numbers mean? They need to be explained in the text.

Response 20: The numbers "23", "25", "213", etc. in Figure 4 mean the band name corresponding to the vegetation index of the UAV or GF-6 data combination, such as NDI23 of the UAV data, which represents the combined vegetation index consisting of band 2 and band 3. The meaning of these figures is given at Table 2, and for the convenience of the readers, the authors have made remarks under Figure 4.

### Discussions

Point 21: Line 546: What is the multisource data? Which results show the multisource data outcomes?

Response 21: In the original manuscript, the authors used the term "multisource data" to summarize both airborne UAV and satellite-based GF-6 data to make it more clear and easy to read, the authors have rewritten it as "UAV and GF-6 data".

Point 22: Lines 562-570: These lines might be better placed in the introduction section.

Response 22: The author has rewritten this section. (See Lines: 606-623).

### Conclusions

Point 23: Lines 605-607: Where are the results showing the combination of UAV and GF-6 data?

Response 23: The author wanted to express the meaning of combining UAV data with MLR (machine learning regression algorithm) and GF-6 satellite data with MLR respectively. the author has corrected these lines. (See Lines: 638-640).

Combining UAV and GF-6 satellite data can simultaneously obtain high-resolution and larger area data to make up for the deficiencies of satellite and airborne data, and the authors will further explore the applicability of combining UAV and GF-6 data in estimating canopy chlorophyll content or other vegetation parameters in their future research careers.
